# Language Models Speed Up Local Search for Finding Programmatic Policies

**Quazi Asif Sadmine**                                                  *sadmine@ualberta.ca*
*Amii, Department of Computing Science*
*University of Alberta*

**Hendrik Baier**                                                       *h.j.s.baier@tue.nl*
*Eindhoven University of Technology*

**Levi H. S. Lelis**                                                    *levi.lelis@ualberta.ca*
*Amii, Department of Computing Science*
*University of Alberta*

**Reviewed on OpenReview:** *https://openreview.net/forum?id=tBkj2I1mJY*

## Abstract

Encoding policies that solve sequential decision-making problems as programs offers advantages over neural representations, such as interpretability and modifiability of the policies. On the downside, programmatic policies are elusive because their generation requires one to search in spaces of programs that are often discontinuous. In this paper, we leverage the ability of large language models (LLMs) to write computer programs to speed up the synthesis of programmatic policies. We use an LLM to provide initial candidates for the policy, which are then improved by local search. Empirical results in three problems that are challenging for programmatic representations show that LLMs can speed up local search and facilitate the synthesis of policies. We conjecture that LLMs are effective in this setting because we give them access to the outcomes of the policies rollouts. That way, LLMs can try policies encoding different behaviors, once they observe what a previous policy has accomplished. This process forces the search to explore different parts of the space through "exploratory initial programs". Experiments also show that much of the knowledge LLMs leverage comes from the domain-specific language that defines the search space - the overall performance of the system drops sharply if we change the name of the functions used in the language to meaningless names. Since our system only queries the LLM in the first step of the search, it offers an economical method for using LLMs to guide the synthesis of policies.

## 1  Introduction

There is a growing interest in using programmatic representations to encode solutions to sequential decision-making problems (Bonet et al., 2010; Aguas et al., 2018; Bastani et al., 2018; Trivedi et al., 2021; Moraes et al., 2023; Liang et al., 2023). Depending on the language used, programs that encode programmatic solutions can generalize to unseen scenarios better than neural representations (Inala et al., 2020; Trivedi et al., 2021). Such solutions can also be interpretable (Verma et al., 2018a; Medeiros et al., 2022; Aleixo & Lelis, 2023), which could allow users to manually modify computer-generated solutions.

The challenge of using programmatic representations is that one needs to search in large and discontinuous spaces of programs, typically defined by a domain-specific language. In the context of supervised learning, a common approach to finding programmatic solutions is to learn a function that guides the search in the programmatic space (Balog et al., 2017; Odena et al., 2021; Odena & Sutton, 2020; Fijalkow et al., 2022; Ameen & Lelis, 2023). This function is learned in a self-supervised manner by exploiting the structure of

the language. For example, in program synthesis, one is given a set of input-output pairs and must find a program that maps every input to its output. A guiding function can be learned by generating a set of random programs from the language. Training problems can be created by providing random input values to these programs, which produce the target outputs. The guiding function is trained to solve the problems represented by the input-outputs pairs whose solution is the program that was randomly generated.

However, it is unclear how to generalize this self-supervised approach to learning guiding functions to solve sequential decision-making problems, where the agent learns by interacting with the environment. This is because, in many settings, the agent does not know the problem it will attempt to solve prior to learning, making it difficult to learn a guiding function as is done in the supervised learning setting. Instead of attempting to learn a guiding function, in this paper, we present a system that leverages the ability of large language models (LLM) to write computer programs encoding "general knowledge" to guide the search for programmatic policies. The type of guidance that we will explore is through the initialization of the search for a program encoding a policy to solve the problem. Similarly to how weight initialization plays an important role in training neural networks (Glorot & Bengio, 2010; LeCun et al., 2012), we show that LLM can also play an important role in initializing the learning process of programmatic policies.

A common approach to synthesizing programmatic policies is to use stochastic local search algorithms to search in syntax-based (Koza, 1992; Carvalho et al., 2024) or semantic-based (Moraes & Lelis, 2024) spaces of programs. The system we introduce in this paper, Local Search with LLM (LS-LLM), proposes a novel integration of an LLM into local search to speed up this process. We hypothesized that an LLM can guide the search for policies by initializing the search in promising parts of the space.

We evaluated our hypothesis on three deterministic domains that are challenging for current systems that generate programmatic policies: Poachers & Rangers (PR) and Climbing Monkey (CM) (Moraes et al., 2023), and MicroRTS (Ontañón, 2013). LS-LLM was more sample efficient than the evaluated baselines, including a system that is identical to LS-LLM but does not use an LLM to initialize the search. We also show that LS-LLM's performance drops sharply once we encrypt the name of the functions and terminal symbols used in the domain-specific language, suggesting that the knowledge the LLM leverages largely comes from interpreting those names. Our results suggest that LLMs can be an economical and effective solution in guiding local search algorithms in synthesizing programmatic policies.[1]

## 2 Problem Definition

Let $(S, A, p, r, \mu, \gamma)$ be a Markov decision process (MDP). Here, $S$ is the set of states, and $A$ is the set of actions. The function $p(s_{t+1}|s_t, a_t)$ encodes the transition function, which returns the probability of reaching $s_{t+1}$ given that the agent is $s_t$ and chooses $a_t$. The agent observes a reward value $R_{t+1}$, which is given by a function $r$, when moving from $s_t$ to $s_{t+1}$. $\mu$ is the distribution of initial states of the MDP and $\gamma$ in $[0, 1]$ is the discount factor. A policy $\pi$ is a function that receives a state $s$ and an action $a$ and returns the probability in which $a$ is chosen in $s$. A solution to an MDP is a policy $\pi$ that maximizes the expected sum of discounted rewards for $\pi$ starting at $s_0 \sim \mu$: $R \doteq \mathbb{E}_{\pi, p, \mu}[\sum_{k=0}^{\infty} \gamma^k R_{k+1}]$.

We consider policies encoded in programs written in a domain-specific language (DSL). The set of programs that a DSL accepts defines the set of policies that the agent can consider. This programmatic policy space is defined as a context-free grammar $(M, \Omega, W, I)$, where $M$, $\Omega$, $W$, and $S$ are the sets of non-terminals, terminals, the production rules of the grammar, and the grammar's initial symbol, respectively. Figure 1 shows an example of a DSL, where $M = \{I, C, B\}$, $\Omega = \{c_1, c_2, b_1, b_2, \text{if}, \text{then}\}$, $W$ are the production rules (e.g., $C \to CC$), and $I$ is the initial symbol.

We represent programs as abstract syntax trees (AST), where each node $n$ and its children represent a production rule if $n$ represents a non-terminal symbol. For example, the root of the tree in Figure 1, which represents the non-terminal $I$, and its children correspond to the production rule $I \to \text{if}(B) \text{ then } C$. Leaf nodes are terminals. Figure 1 shows an example of an AST for "if $b_1$ then $c_1\ c_2$". A DSL $D$ defines the possibly infinite space of programs $[\![D]\!]$; in our case, each $p$ in $[\![D]\!]$ represents a policy. Given such a space,

---

[1]The source code used to run the experiments of this paper is available online: **https://github.com/asifsadmain/LS-LLM**

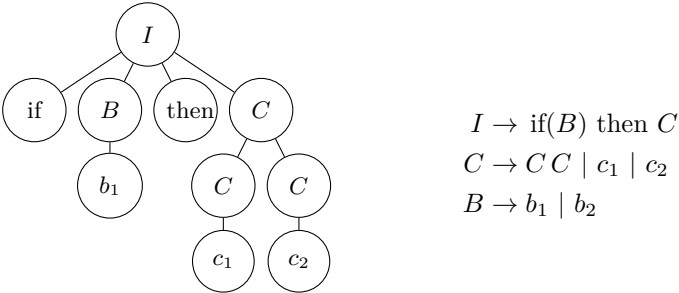

$$I \rightarrow \text{if}(B) \text{ then } C$$
$$C \rightarrow C\,C \mid c_1 \mid c_2$$
$$B \rightarrow b_1 \mid b_2$$

Figure 1: Abstract syntax tree for "if $b_1$ then $c_1$ $c_2$" and the language in which the program is written.

we need to find a policy that approximates a solution to $\max_{\pi \in [\![D]\!]} \mathbb{E}_{\pi,p,\mu}[\sum_{k=0}^{\infty} \gamma^k R_{k+1}]$. We use a local search algorithm to find an approximate solution to this optimization problem (Carvalho et al., 2024).

## 3    Programmatic Policy Learning as Local Search

Previous work showed that a simple implementation of stochastic hill-climbing (SHC) achieves state-of-the-art results in challenging domains (Carvalho et al., 2024; Moraes & Lelis, 2024), so this is what we use in our experiments. SHC starts with an arbitrary policy $\pi'$ as a candidate that maximizes the agent's return.

The initial candidate is generated starting with the initial symbol $I$ and applying a production rule to replace $I$; the rule is chosen uniformly at random or according to a probability distribution, if one is available (Trivedi et al., 2021). We replace the leftmost non-terminal $X$ of the resulting string with the symbols on the right-hand side of a production rule for $X$; the rule is randomly chosen among those for $X$. This process continues until a string with only terminal symbols is generated, which represents the search's initial candidate.

SHC searches in the space defined by both the DSL and a neighborhood function $\mathcal{N}_k(\pi)$. This function receives a candidate $\pi$ and returns a set of $k$ policies—the neighbors of $\pi$ in the search space. SHC evaluates all $k$ neighbors in terms of an estimate of the return $R$ of each neighbor; SHC selects the neighbor of $\pi$ with the largest estimated value $R$. The value of $R$ is estimated by rolling out each of the neighboring policies. This process is repeated with the newly selected candidate policy. SHC stops if none of the neighbors has an estimated $R$-value that is larger than the current candidate; the search reaches a local optimum. We implement SHC with a restarting strategy: once SHC reaches a local optimum, we restart the search from a randomly chosen initial candidate. SHC with restarts returns the best solution encountered in all searches.

SHC is stochastic because the generation of the initial candidate and the neighborhood function are stochastic. In non-deterministic environments, we can only approximate the values of $R$ when deciding which neighbor to select. The neighbors $\mathcal{N}_k$ returns are generated as follows. Each of the $k$ neighbors is generated by randomly choosing a node $n$ in the AST of the program $\pi$ that represents a non-terminal symbol (e.g., the nodes $I$, $B$ or $C$ in Figure 1) and replacing the sub-tree rooted at $n$ with a randomly generated sub-tree. This new sub-tree is generated by following the same process described to generate the initial candidate. The difference is that we start by replacing the non-terminal of node $n$, instead of the initial symbol $I$.

## 4    Local Search with LLM (LS-LLM)

Local Search with LLM (LS-LLM) is described in Algorithm 1, and a schematic view of it is shown in Figure 2. It receives the MDP $\mathcal{P}$, a large language model $M$, a budget $B$ that specifies the number of model queries that are allowed, a domain-specific language $D$, and a local search algorithm, SHC in our implementation. LS-LLM returns an approximate solution to $\mathcal{P}$. Figure 2 shows which code lines in Algorithm 1 interact with the MDP (first column), the LLM (second column), and SHC (third column).

LS-LLM leverages the ability of the LLM to encode common knowledge from its training data, including the ability to generate computer programs, to speed up the search for programmatic policies. LS-LLM first requests from the LLM a set of policies that solve the MDP, for which a textual description is provided

---

**Algorithm 1** LS-LLM

---

**Require:** MDP $\mathcal{P}$, large language model $M$, model budget $B$, domain-specific language $D$, search algorithm SHC.
**Ensure:** An approximate solution $\pi$ to $\mathcal{P}$.

 1: $\pi \leftarrow$ ask for a policy that solves $\mathcal{P}$ written in $D$ from $M$
 2: $\mathcal{F} \leftarrow$ evaluate $\pi$ in $\mathcal{P}$ and return feedback
 3: **for** $i = 1, \cdots, B - 1$ **do**
 4: $\quad \pi' \leftarrow$ ask for a policy that solves $\mathcal{P}$ written in $D$ from $M$ while providing $\mathcal{F}$ as input to $M$
 5: $\quad \mathcal{F}' \leftarrow$ evaluate $\pi'$ in $\mathcal{P}$ and return feedback
 6: $\quad$ **if** $\pi'$ is an improvement over $\pi$ in terms of return $R$ **then**
 7: $\quad\quad \pi \leftarrow \pi'$
 8: $\quad\quad \mathcal{F} \leftarrow \mathcal{F}'$
 9: **return** the result of SHC while using $\pi$ as initial candidate

---

(lines 1 and 4). The policies are requested sequentially from the model. In this way, LS-LLM can provide feedback to the LLM by rolling out the policy in the environment (e.g., by informing the model of the actions the agent has taken in the environment); the feedback is denoted as $\mathcal{F}$ in Algorithm 1. Note that LS-LLM does not receive any feedback when requesting the first policy from $M$. In this case, the prompt simply requests a "strong" policy for the MDP (see Section 5.5 and the supplementary materials for details).

The best policy, in terms of the average return $R$, that the LLM generates is given as input to SHC, which attempts to further improve the policy with search (line 9). Even if the LLM is unable to directly generate a strong policy, its attempt at generating one might allow the search to start in a more promising part of the programmatic space. Such an initialization might allow for quick improvements with SHC.

## 4.1 LS-LLM in Multi-Agent Settings

In our experiments, we consider MDPs given by two-player zero-sum games. In this section, we explain Algorithm 1 for this setting. In this setting, instead of approximating a solution to one MDP, we need to approximate a solution to multiple MDPs. Each MDP is defined by fixing the opponent, and searching for a policy that maximizes the agent's return for that MDP; the opponent can be seen as part of the environment. In game theory terms, by maximizing the agent's return, we approximate a best response to the opponent. We use self-play algorithms to define the MDPs to be solved so that the agent can learn to play a game.

Iterated Best Response (IBR) is the simplest of such self-play algorithms. Let $i$ and $-i$ be the two players. IBR starts with an arbitrary policy $\pi_0$ in $[\![D]\!]$ for one of the players, for example, $-i$, and approximates a best response $\pi_1$ to $\pi_0$ for $i$. Then, it approximates a best response to $\pi_1$ for $-i$. This process is repeated $n$ times, which is determined by a computational budget. The last policies, $\pi_n$ and $\pi_{n-1}$, are returned as an approximate solution to the game. In IBR, we use Algorithm 1 to compute each of these best responses.

The self-play process IBR follows generates a sequence of policies for $i$ and $-i$, but IBR only considers the latest policy while computing a best response, which can hinder the algorithm's ability to learn how to play the game. Other algorithms, such as Fictitious Play (FP) (Brown, 1951), are more informative in guiding the learning process. This is because FP computes best responses to a policy that mixes all the best responses computed in previous iterations. Intuitively, FP forces the agent to learn to play against all of its previous versions. In practical terms, when evaluating the return $R$ of a policy in Algorithm 1, we evaluate the policy's return on multiple MDPs, one for each best response computed in the FP process. If FP computed the policies $\pi_1, \pi_2, \cdots, \pi_{t-1}$ so far, then the average return of policy $\pi_t$ in Algorithm 1 is $\sum_{j=1}^{t-1} \frac{1}{t-1} R_j$. Here, $R_j$ is the return of $\pi_t$ in the MDP in which $\pi_j$ is fixed as part of the environment.

Other self-play algorithms, such as Double Oracle (DO) (McMahan et al., 2003), can be selective with which opponents to fix and fold into the environment. In particular, Local Learner (2L) (Moraes et al., 2023) includes only the policies that were shown to provide search guidance in the program space. As a result, the evaluation of $R$ for both DO and 2L can be computationally cheaper than the evaluation of $R$ if FP is used. This is because the evaluation can be performed in fewer MDPs for DO and 2L. Previous work showed that

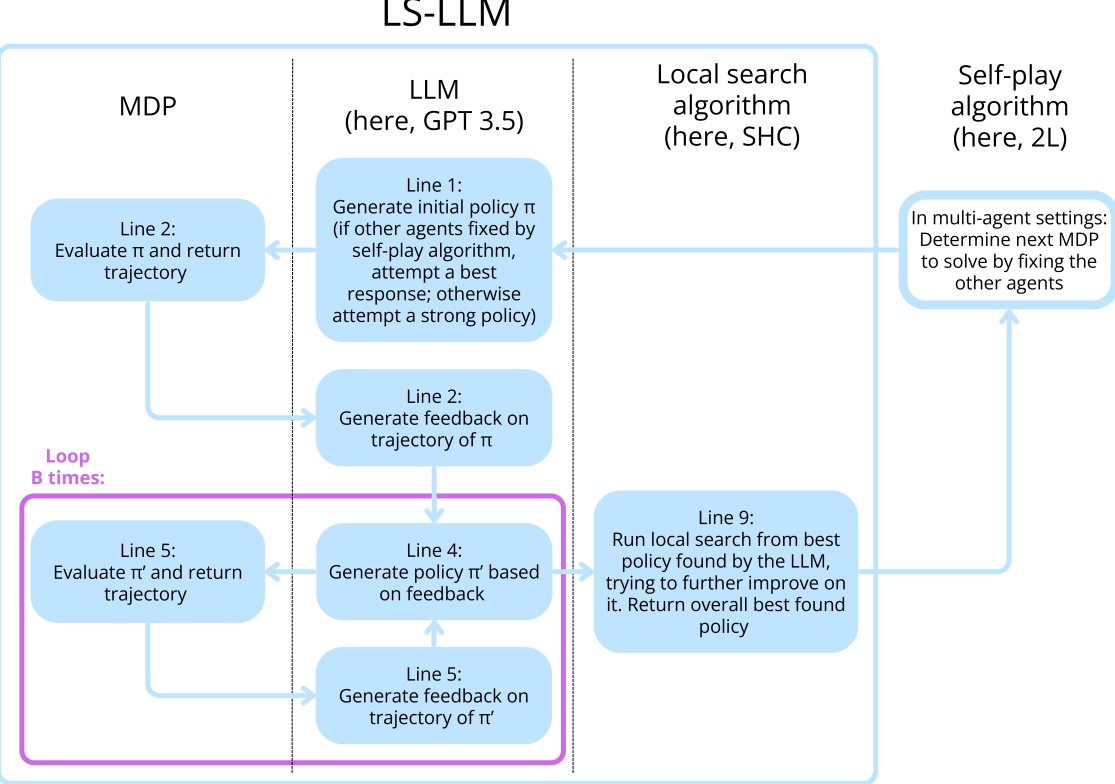

Figure 2: Schematic view of Algorithm 1. This scheme shows LS-LLM's interactions with the MDP (first column) and calls to the LLM (second column) and calls to the local search algorithm (third column). The scheme also shows the relationship between LS-LLM and the self-play algorithm, 2L.

FP, DO, and 2L tend to perform better than IBR in the context of programmatic policies (Moraes et al., 2023). Independently of the learning algorithm used to choose the MDPs, the basic repeated operation of these algorithms is to find a programmatic policy that maximizes the agent's return in a given MDP. In this paper, we consider LS-LLM to approximate solutions to MDPs within learning algorithms such as 2L.

## 5 Empirical Methodology

We evaluate in three games our hypothesis that LLMs can be used to speed up the synthesis of programmatic policies by seeding the SHC search: Poachers & Rangers (PR) and Climbing Monkey (CM) (Moraes et al., 2023), and MicroRTS (Ontañón et al., 2018). PR and CM are challenging for programmatic policies but easy for humans; MicroRTS is a challenging real-time strategy game with an annual competition.[2]

### 5.1 Poachers & Rangers

PR is an unconstrained security game in which one of the players tries to defend a national park (rangers) and the other player attempts to attack the park (poachers). Given a number of gates $N$, rangers win the game if they defend all attacked gates; poachers win the game if they attack an unprotected gate. The game is unconstrained because rangers can choose to defend all $N$ gates and poachers can choose to attack all $N$ gates. Programs encoding strategies for rangers are of the form `defend(1)`, `defend(20)`, while encoding strategies for poachers are of the form `attack(1)`, `attack(2)`, `attack(10)`. Although the optimal strategy for rangers is to trivially defend all gates, the synthesis of this strategy through self-play was shown to be difficult (Moraes et al., 2023). This is because the program can be arbitrarily long, depending on $N$, and

---

[2]https://sites.google.com/site/micrortsaicompetition/

the algorithm needs to discover that each of the gates must be defended. The DSL used in this domain does not allow for loops, which could dramatically simplify the task. We use this domain with a DSL without loops in our experiments because it represents a challenging problem for synthesizers performing search, but it can potentially be simple for systems that leverage general-knowledge systems such as LLMs.

## 5.2 Climbing Monkey

CM is a game in which two monkeys compete in a climbing contest: the monkey that can reach the highest branch of a tree is the winner; the game ends in a draw if both monkeys climb to the same branch. In this game, a monkey must climb to branch $i + 1$ from $i$; no branch can be skipped. For example, the strategy `climb(1)`, `climb(2)`, `climb(20)` allows the monkey to reach the branch 2 of the tree; instruction `climb(20)` is ignored. Similarly to PR, the optimal strategy is trivial, but hard to achieve with current systems. In contrast to PR, the instructions in the optimal program must be ordered from branch 1 to $N$, which presents different challenges to the algorithms. Similarly to PR, the DSL for CM also does not allow for loops.

For both PR and CM we provide as feedback $\mathcal{F}$ a textual description of the end-game state of the match played between $\pi_t$ and the last policy generated by the self-play algorithm, $\pi_{t-1}$. For PR, the feedback describes which gates were protected and which undefended gates the poachers attacked. For CM, the feedback reports how many branches the monkeys climbed and if they won or lost the match. The prompts with the feedback are given in Appendix C.

## 5.3 MicroRTS

MicroRTS is a challenging real-time strategy game in which the agent controls many units in real time. The game is played on a gridded map that can vary in structure and size; different maps often require different strategies. In MicroRTS, the player starts with a Base unit and, depending on the map, with other units. The Base allows the player to train Worker units and store resources; Workers can build structures (Base or Barracks), collect resources, and attack opponent units. Barracks can train combat units, including Light, Ranged, and Heavy units. These units differ in their resource cost, the amount of damage they can cause to other units, the amount of damage they can suffer before being removed from the game, and in attack range. A player wins the game if they eliminate all units and structures of the other player. We use the following maps from the MicroRTS repository,[3] with the map size in brackets: NoWhereToRun ($9 \times 8$), DoubleGame ($24 \times 24$), and BWDistantResources ($32 \times 32$). The images of the maps are provided in Appendix A.

The MicroRTS policies are written in Microlanguage, a DSL of the game (Mariño et al., 2021). The Microlanguage includes functions such as `harvest` and `moveToUnit`, which allow one to encode strong policies even in short programs. The Microlanguage uses for-loops to prioritize actions. That is, functions called earlier in a loop have a higher priority over those called later. The Microlanguage is described in Appendix C.

MicroRTS is not a symmetric game depending on the starting location of the players. If policy $\pi_1$ defeats $\pi_2$ when the former starts at location 1 and the latter at location 2, it does not mean that $\pi_1$ will defeat $\pi_2$ if we swap their locations. To ensure a fair evaluation, each pair of policies plays two matches on each map, one in each initial location.

We used GPT 3.5, which was trained with data up to September 2021. To the best of our knowledge, no programmatic policy for PR, CM, and MicroRTS had been published online before the model's training cut-off date. The repository of Mariño et al. (2021) contains programs written in an earlier version of Microlanguage. This earlier version of the language differs significantly from the version we use in our experiments in terms of the domain-specific functions available and the number of parameters required in the functions. To illustrate, none of the programs available in Mariño et al.'s repository can be run in the interpreter of our version of the language. Moreover, Mariño et al. did not experiment with the NoWhereToRun and BWDistantResources maps that we use. Considering the differences in the language and in the maps used, it is unlikely that Mariño et al.'s programs have influenced our results.

---

[3] https://github.com/Farama-Foundation/MicroRTS/

### 5.4 Baseline Systems

We evaluate LS-LLM with 2L as the learning algorithm; that is, we use 2L to determine the sequence of MDPs LS-LLM is invoked to synthesize policies for. In the results plots, we call this combination 2L(LS-LLM). We use 2L because it was shown to outperform IBR, FP, and DO in the three games that we use in our experiments (Moraes et al., 2023). Furthermore, 2L placed second in the 2023 MicroRTS competition, only behind a programmatic strategy written by human programmers (MicroRTS AI Competition, 2023). Therefore, 2L(LS) (using local search, but without LLM) is our main baseline. We also use FP(LS) and IBR(LS) as baselines to offer a larger pool of policies. All algorithms used the same SHC implementation. Note that in self-play algorithms, we need to compute a sequence of best-responses—i.e., solve a sequence of MDPs. While 2L(LS-LLM) uses an LLM to initialize each of these searches, the baselines 2L(LS), FP(LS), and IBR(LS) use the latest best response computed to initialize the search in the programmatic space.

We did not consider deep reinforcement learning baselines because we focus on testing our hypothesis that LLMs can be used to speed up the synthesis of programmatic policies. Moreover, the 2023 MicroRTS competition showed the general trend that one should expect from such a comparison. On smaller maps, a neural policy outperforms programmatic ones. This is because the DSL used in the competition and also in our experiments does not allow for fine-grained control of the game units, which neural policies can quickly learn on smaller maps. However, as the size of the maps increases, programmatic policies outperform neural ones. This is because the inductive bias the DSL provides allows for the synthesis of policies with strong high-level strategies (e.g., how to train stronger units and reach the opponent in larger spaces). See the MicroRTS AI Competition (2023) website for more information.

We also did not consider other programmatic representations such as decision trees (Bastani et al., 2018) and finite state machines (Koul et al., 2019) in our experiments because they either require one to first train a neural model that is then distilled into a programmatic representation or to directly map the neural model onto a program (Orfanos & Lelis, 2023). Due to this dependency on using a neural policy to guide the synthesis process, it is unclear how to leverage LLMs to seed up learning with these representations. One could use decision trees and finite state machines, without neural guidance, as the underlying representation of the policies. However, we would lose the inductive bias the domain-specific language provides while still paying the cost of having to search in discrete and discontinuous spaces of programs.

### 5.5 Language Model and Prompts

The self-play algorithms require an initial policy to start the iterative process of computing best responses. For 2L(LS-LLM), we request an initial strategy from the LLM. For all baselines, we randomly generate this initial policy, as described in Section 3. 2L(LS-LLM) uses three different prompts for each domain: one for the initial policy of the first iteration of 2L (called "initial"), one for the model's first attempt to generate a best response to $\pi_t$ (called "first-attempt"), and one for the remaining $B - 1$ attempts (called "feedback-attempt"). We provide a description of the domain-specific language in all three prompts. The description is given as a context-free grammar with explanations of the functions used in the language. In first-attempt and feedback-attempt we also provide the last policy generated in self-play, $\pi_t$, and explain that this is the policy that needs to be defeated. In the feedback-attempt we provide a sample of 10 actions of the model's previous attempt at a best response to $\pi_t$ issued in a match against $\pi_t$. We do not provide all actions issued in the match due to the model's limited context length. We provide all prompts in Appendix C.

### 5.6 Experiments Performed

We performed three experiments. In the first experiment, we compare 2L(LS-LLM) on PR, CM, and the three MicroRTS maps to our baselines 2L(LS), FP(LS), and IBR(LS). These baselines use the same local search algorithm as LS-LLM, SHC, but without LLMs. In this experiment, we compare the different approaches in terms of gates protected (PR), branches climbed (CM), and winning rate (MicroRTS). The winning rate of a policy is computed for a set of opponent policies and is computed while using as opponents the policies the other evaluated systems generate. Specifically, we sum the number of victories and half the number of draws and divide this sum by the total number of matches played (Ontañón, 2017). For example,

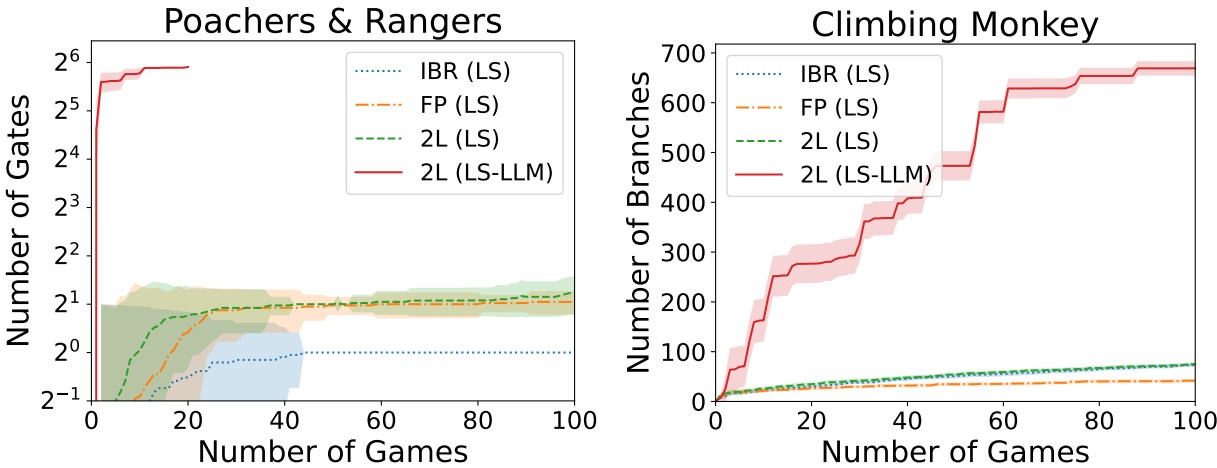

Figure 3: Average number of gates covered and branches climbed by the number of games played for 2L(LS-LLM) (ours), 2L(LS), FP(LS), and IBR(LS) in the games of PR and CM. The average and the 95% confidence interval are over 30 independent runs of the systems.

the winning rate of a strategy that wins 5, loses 2, and draws 3 matches is $\frac{5+1.5}{10} = 0.65$. The performance metric of each domain is evaluated in terms of the number of games each system needs to play to achieve a level of performance. The results are presented in plots where the y-axis shows the performance and the x-axis the number of games played (Figure 3 shows two examples). This experiment evaluates our hypothesis that LS-LLM can speed up the process of learning policies, in terms of the number of games played.

The second and third experiments are intended to improve our understanding of LS-LLM. In the second, we evaluate LS-LLM while removing from feedback-attempt the set of actions sampled from the match played between the LLM-generated policy and the last policy generated by the self-play algorithm. This experiment measures the impact of the feedback on the generation of an initial candidate policy for the SHC search.

In the third experiment, we "encrypt" the names of the functions and nonterminal symbols in the Microlanguage. We still explain in the prompt what each function does, but we use meaningless names for them. For example, in the prompts used in the first and second experiments, we explain that the function `hasNumberOfUnits(T, N)` *checks if the ally player has N units of type T*. In the third experiment, this function is called `b1`, with the same explanation provided. Our goal is to verify how important the names of functions and nonterminals are to the LLM.

### 5.6.1 Other Specifications

All experiments were run on computers with 2.6 GHz CPUs and 12 GB of RAM. We used $B = 5$ for LS-LLM. In PR, we set the number of gates to 60 and leave the number of branches for CM unbounded. We use the value of $k = 1,000$ in $\mathcal{N}_k$. SHC is run with a time limit of $2,000$ seconds. Once SHC reaches a local optimum, if there is still time allowed to search, it restarts the search from the initial candidate the LLM has suggested. After reaching the time limit, the search returns the best program encountered in all runs. This is SHC's approximation to a solution to the MDP.

## 6 Empirical Results

The results of our first experiment are shown in Section 6.1 ("Synthesizing Programmatic Policies") and of the second and third experiments in Section 6.2 ("Ablation Experiments").

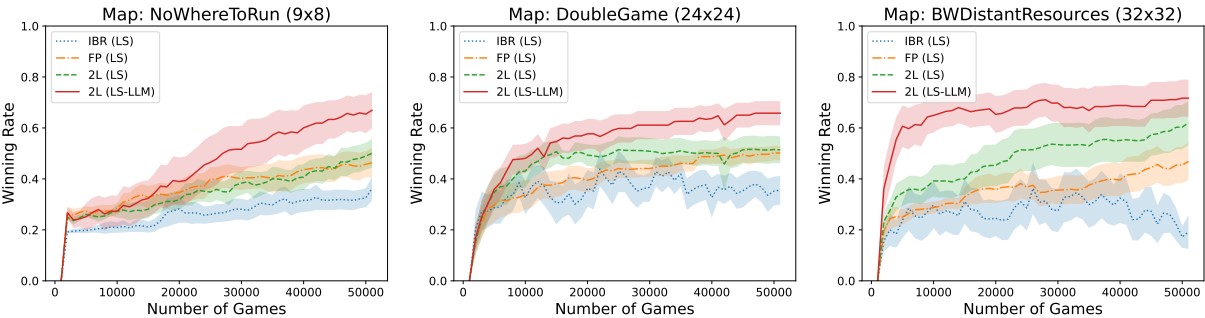

Figure 4: Average winning rate by the number of games played for 2L(LS-LLM) (ours), 2L(LS), FP(LS), and IBR(LS) in MicroRTS. The winning rate is computed by having the strategy a system synthesized, at a given number of games played, play against the strategies each of the other systems synthesized after the maximum number of games was played (50,000). The average and the 95% confidence interval are over 30 independent runs of the systems.

## 6.1 Synthesizing Programmatic Policies

Figure 3 presents the results for PR and CM. 2L(LS-LLM) represents our contribution, which uses 2L as the learning algorithm, and LS-LLM to approximate the best responses. IBR(LS), FP(LS), and 2L(LS) are our baselines. We refer to the algorithms in the abbreviated forms LS-LLM, IBR, FP, and 2L.

LS-LLM outperforms all baselines by a large margin. In PR, LS-LLM learns to defend all 60 gates with less than 20 games played. Often, the initial policy the LLM suggests already covers most of the gates in the game. By contrast, progress for systems that rely only on search is slow. The LLM not only generates best responses, but the best responses it generates cover more gates than is required to best respond to the poachers policy. For example, if the current poachers policy $\pi_p$ is `attack(1)`, then the simplest best response to it is `defend(1)`, which is the response an algorithm searching in the space of programs will most likely find, due to the minimal size of the program. Instead, the LLM often returns a best response such as `defend(1)`, `defend(2)`, `defend(3)`, which allows quicker progress toward the policy covering all 60 gates.

We observe a similar behavior for the CM domain, where LS-LLM quickly generates programs that climb to much higher branches than the programs the baselines generate. Most of the progress that we observe for LS-LLM comes from the best responses the LLM suggests. As programs become longer, it becomes less likely that a neighbor program will represent an improvement with respect to the current program.

Figure 4 shows the results for MicroRTS, where the winning rate is computed by having the policy a system generated, for a given number of games played, play against the policy each of the other systems generated after the maximum number of games used in the experiment (50,000). LS-LLM outperforms all baselines by a large margin in the three maps. Especially for the larger maps, 24×24 and 32×32, LS-LLM generates stronger policies more quickly than the baselines. For example, 2L needs to play approximately 10 times more games than LS-LLM on the 32×32 map to reach the winning rate around 60%.

Although the results on MicroRTS are similar to those in PR and CM, the explanation why LS-LLM performs better than the baselines is different in MicroRTS. In PR and CM, the LLM is often able to generate a program that already solves the MDP. In MicroRTS this happens only in the early iterations of learning, when the self-play policies are weak. Later, as the system generates stronger policies, the LLM fails to directly generate an optimal policy. LS-LLM is more sample-efficient in MicroRTS due to its initialization of the search in the neighborhood of policies that can solve the MDP. We conjecture that the LLM helps the search explore different types of policy. For example, the best policies in the 9×8 map require the use of Ranged units. The LLM quickly generates programs that can train various types of unit, including Ranged units. The process of discovering such policies with search alone takes more iterations. As representative examples, we show in Appendix B two examples of policies the LLM generated, as its attempt to provide a best response to a target policy. The two policies encode different types of strategies for playing MicroRTS.

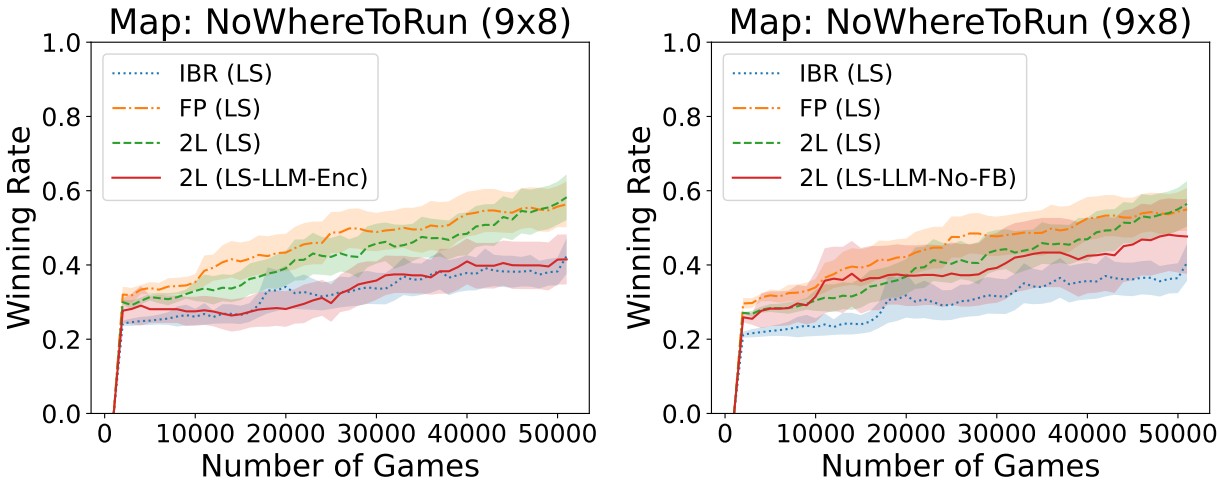

Figure 5: Average winning rate by the number of games played for 2L(LS-LLM-Enc) (left) and 2L(LS-LLM-No-FB) (right) in MicroRTS. The average and the 95% confidence interval are over 30 independent runs of the systems.

## 6.2 Ablation Experiments

Figure 5 shows the results of the second (left) and third (right) experiments. 2L(LS-LLM-Enc) is the version of LS-LLM where we "encrypt" the name of the functions of the DSL. 2L(LS-LLM-Enc) performs similarly to IBR and is worse than FP and 2L. These results suggest that much of the general knowledge that the LLM leverages to provide initial candidates for search comes from the function names.

2L(LS-LLM-No-FB) is the version of LS-LLM that does not provide feedback to the LLM. 2L(LS-LLM-No-FB) performs comparable to the baselines, suggesting that the feedback encodes valuable information. We conjecture that the feedback allows the LLM to verify whether the program it generates achieves the goals the LLM sets to it. We make such a conjecture because the LLM often explains why it generated a policy, as it outputs the policy. For example, it could state that its policy will train Ranged units to protect the Base. The feedback could show that no Ranged units are being trained, allowing the LLM to modify its policy to achieve its goal in its next attempt at generating a programmatic best response.

## 7 Related Work

Programmatic reinforcement learning is an active area of research (Verma et al., 2018b; 2019; Inala et al., 2020; Trivedi et al., 2021; Qiu & Zhu, 2022; Carvalho et al., 2024). All of these previous works employ a form of search in programmatic spaces to solve MDPs. An SMT solver has been used to synthesize policies for single-player puzzles (Butler et al., 2017) and a SAT solver has been used for logic games (Farzan & Kincaid, 2018). We differ from all these previous works in that we use an LLM to speed up the search for programmatic policies.

Others have considered multi-agent settings similar to the ones used in our experiments, but have mainly aimed at extracting interpretable policies from neural models (Milani et al., 2022; Liu et al., 2023c). This contrasts with our approach of searching for policies in DSL-defined spaces. Still in the multi-agent setting and closer to our work, various approaches have been used to find policies for two-player zero-sum games, from genetic programming (Mariño & Toledo, 2022), to Monte Carlo Tree Search (Medeiros et al., 2022) and local search (Mariño et al., 2021; Moraes et al., 2023). Previous work has focused on searching for generating policies, but has not been enhanced by LLMs.

Previous work has used LLMs in games, but not for generating programmatic policies. Much of this work has focused on online planning in open-world games such as Minecraft or Sims-like social simulations, where LLMs have been used to perceive, plan, and act (Park et al., 2023), often decomposing long-horizon goals into

subtasks and monitoring their execution (Wang et al., 2023b), and/or integrating additional agent features such as memory (Zhu et al., 2023) and/or automatic learning curricula (Wang et al., 2023a). In contrast to these efforts, our work generates programmatic policies that are not modified or re-planned in an online fashion; we use LLMs to help us generate such policies.

LLMs have also been used to help generate general policies (Celorrio et al., 2019) to solve classical planning problems (Silver et al., 2023). The classical planning setting differs from ours in that it offers a formal description of the problem. This is in contrast to our setting, where only a less structured description of the problem is available. Another area is the generation of code for robot decision-making, which has seen successful use of LLMs. Singh et al. (2023) generate plans for robotic tasks by providing Python-like code as a prompt to the LLM, including an example of a program to solve a task. The LLM then implements a function to solve another task. Liang et al. (2023) follow a similar approach, where instructions of what needs to be done can be provided as comments in the prompt code. These approaches rely on the LLM to directly solve the problem. This contrasts with our work, where we do not assume that the LLM can solve the MDP directly, so we use it to seed the search for a policy that will solve the MDP.

Another view of our work is the use of LLMs to guide local search. LLMs have been used to solve optimization problems with an interactive prompting loop that asks for new solutions based on previous solutions (Yang et al., 2023; Guo et al., 2023); they have been used to guide Monte Carlo tree search by providing a world model and a heuristic (Zhao et al., 2023); and LLMs have been integrated into genetic operators in many algorithms (Lehman et al., 2022; Liu et al., 2023b; Meyerson et al., 2023; Chen et al., 2023), including multi-objective (Liu et al., 2023a) and quality-diversity evolutionary algorithms (Nasir et al., 2023). Most of the previous work is computationally expensive because they use the LLM in every step of the search(Liu et al., 2023a), while we use it to only initialize the search.

Contemporary to our work, Liu et al. (2024) also showed how to improve the sample efficiency of a local search algorithm in the context of programmatic policies. Similarly to our results, Liu et al. also found that LLMs can be effective in initializing the search for programmatic policies. Although the two independently developed works present similar algorithmic solutions, they approach the problem from complementary perspectives. While Liu et al. focus on different programmatic representations (general-purpose versus domain-specific languages), we focus on the feedback we provide to the LLM, so that it can generate better initial candidate solutions. Moreover, the problem domains used in the experiments of the two papers are disjoint, further validating the idea of seeding the programmatic search with LLM-generated programs.

# 8 Discussion and Conclusions

The results in PR, CM, and MicroRTS showed that LLMs can be used to enhance stochastic local search algorithms in the context of programmatic policies, even if the model is used only sparingly, to seed the search in the programmatic space. The LLM model we used in our experiments was not trained with data about PR and CM, since these games were published after the cutoff date to collect training data to train the model. Although the model had some knowledge of MicroRTS, it did not have knowledge of the Microlanguage and the programmatic policies one can write with it. However, the general knowledge encoded in the model allowed it to attempt "reasonable" programs that likely forced the search to better explore the policy space, thus improving the sample efficiency of the overall system.

Programmatic representations of policies can offer advantages over alternative representations, such as interpretability. The drawback of programmatic representations is that one needs to solve hard combinatorial search problems to generate programs encoding policies. Such search problems are particularly challenging because the optimization landscape is full of discontinuities and thus is not suitable for gradient descent optimization. In this paper, we presented LS-LLM, a system that uses an LLM to guide the search for programmatic policies. The LLM is used only to initialize the search in the programmatic space, making LS-LLM a viable method given the often high monetary costs of using LLMs. We hypothesized that the use of an LLM would speed up the process of computing policies in terms of the number of samples required to learn such policies. We tested our hypothesis in three games that are challenging for current systems, including a challenging real-time strategy game. Our results supported our hypothesis, as the programmatic

policies LS-LLM generated were stronger than those generated by current state-of-the-art systems for the same number of played games.

## Acknowledgments

This research was supported by Canada's NSERC and the CIFAR AI Chairs program, and was enabled in part by support provided by the Digital Research Alliance of Canada. This research has also received funding from the project ALIGN4Energy (NWA.1389.20.251) of the research programme NWA ORC 2020 which is (partly) financed by the Dutch Research Council (NWO), and from the European Union's Horizon Europe Research and Innovation Programme, under Grant Agreement number 101120406. The paper reflects only the authors' view and the EC is not responsible for any use that may be made of the information it contains.

The authors thank the reviewers for their helpful suggestions.

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

# A MicroRTS Maps

Figure 6 shows the three MicroRTS maps used in our experiments.

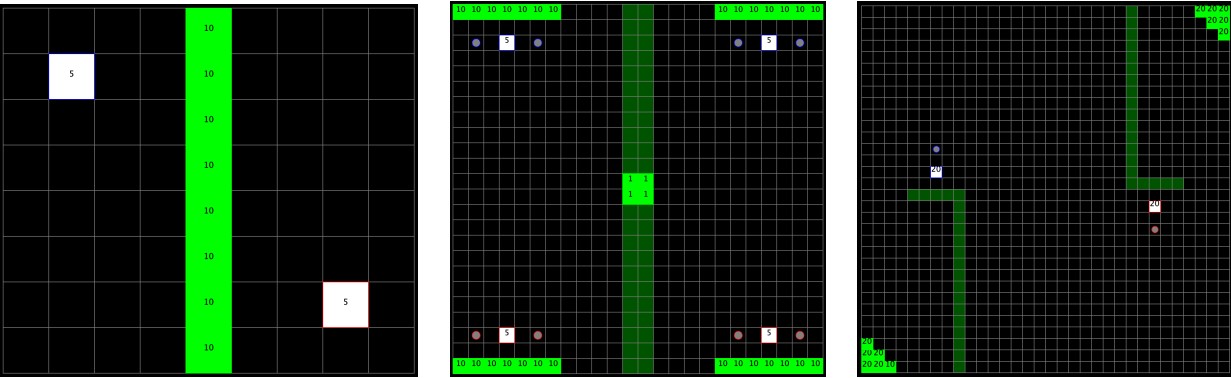

Figure 6: The three maps used in our experiments: NoWhereToRun (9 × 8) on the left; DoubleGame (24 × 24) in the middle; BWDistantResources (32 × 32) on the right.

| Best Response Attempt 1 | Best Response Attempt 2 |
|---|---|
| ```
1  for(Unit u)
2      if(OpponentHasNumberOfUnits(Ranged,5))
3          u.train(Heavy,Up,10)
4      if(OpponentHasNumberOfUnits(Light,10))
5          u.train(Ranged,Up,5)
6          u.train(Heavy,Up,5)
7      if(OpponentHasNumberOfUnits(Worker,5))
8          u.attack(Weakest)
9      u.train(Light,Up,20)
10     u.build(Barracks,Up,1)
11     for(Unit u)
12         u.attack(Closest)
``` | ```
1  for(Unit u)
2      u.build(Barracks,Right,1)
3      u.train(Ranged,Down,10)
4      u.attack(Strongest)
5      u.train(Light,Left,5)
6      u.attack(Closest)
``` |

Table 1: Examples of programmatic best responses the LLM generated to the same target policy given by 2L.

# B  Examples of LLM-Generated Programs

We conjectured in Section 6.1 that LS-LLM is effective because the LLM allows it to quickly explore different types of strategies. Table 1 shows two examples of policies that the LLM generated as attempts to provide a best response to the same target policy provided by the 2L self-play algorithm for the BWDistantResources map. The two policies encode different strategies for playing MicroRTS. The policy on the left decides on the types of unit the player trains based on the units the opponent controls. For example, it will train Heavy units if the opponent controls Ranged units (lines 2 and 3). The policy on the right focuses on training Ranged units (line 3). Later in the game, after the player controls 10 Ranged units, if resources are still available, it will train Light units (line 5). These two policies do not include instructions for collecting resources and training Worker units. Therefore, while they might not be effective in practice, they offer different "template functions" that the local search algorithm can further improve into effective policies.

# C MicroRTS Prompts

## C.1 Initial Attempt (Example for 9x8 Map)

Consider a 9x8 gridded map of MicroRTS, a real-time strategy game. Consider this map as a 2 dimensional array with the following structure:

– There are a total of 8 neutral resource cells situated along the central column of the map, dividing the map into two parts. Each resource cell contains 10 units of resources.

– The base B1 of player 1 is located at index (1,1), which is located on the left side of the map.

– The base B2 of player 2 is located at index (7,6), which is located on the right side of the map.

– Each player controls one base, which initially has 5 units of resources.

– The only unit a player controls at the beginning of the game is the base.

Consider this Context-Free Grammar (CFG) describing a programming language for writing programs encoding strategies of MicroRTS. The CFG is shown in the <CFG></CFG> tag bellow:

<CFG>

$S \rightarrow SS$ | for(Unit u) S | if(B) then S
| if(B) then S else S | $C$ | $\lambda$

$B \rightarrow u.\text{hasNumberOfUnits}(T, N)$
| $u.\text{opponentHasNumberOfUnits}(T, N)$
| $u.\text{hasLessNumberOfUnits}(T, N)$
| $u.\text{haveQtdUnitsAttacking}(N)$
| $u.\text{hasUnitWithinDistanceFromOpponent}(N)$
| $u.\text{hasNumberOfWorkersHarvesting}(N)$
| $u.\text{is\_Type}(T)$ | $u.\text{isBuilder}()$
| $u.\text{canAttack}()$ | $u.\text{hasUnitThatKillsInOneAttack}()$
| $u.\text{opponentHasUnitThatKillsUnitInOneAttack}()$
| $u.\text{hasUnitInOpponentRange}()$
| $u.\text{opponentHasUnitInPlayerRange}()$
| $u.\text{canHarvest}()$

$C \rightarrow u.\text{build}(T, D, N)$ | $u.\text{train}(T, D, N)$ | $u.\text{moveToUnit}(T_p, O_p)$
| $u.\text{attack}(O_p)$ | $u.\text{harvest}(N)$
| $u.\text{attackIfInRange}()$ | $u.\text{moveAway}()$

$T \rightarrow \text{Base}$ | Barracks | Ranged | Heavy
| Light | Worker

$N \rightarrow 0$ | 1 | 2 | 3 | 4 | 5 | 6 | 7 | 8 | 9
| 10 | 15 | 20 | 25 | 50 | 100

$D \rightarrow \text{EnemyDir}$ | Up | Down | Right | Left

$O_p \rightarrow \text{Strongest}$ | Weakest | Closest | Farthest
| LessHealthy | MostHealthy | Random

$T_p \rightarrow \text{Ally}$ | Enemy

</CFG>

This language allows nested loops and conditionals. It contains several Boolean functions (B) and command-oriented functions (C) that provide either information about the current state of the game or commands for the ally units.

The Boolean functions ('B' in the CFG) are described below:

1. u.hasNumberOfUnits(T, N): Checks if the ally player has N units of type T.

2. u.opponentHasNumberOfUnits(T, N): Checks if the opponent player has N units of type T.

3. u.hasLessNumberOfUnits(T, N): Checks if the ally player has less than N units of type T.

4. u.haveQtdUnitsAttacking(N): Checks if the ally player has N units attacking the opponent.

5. u.hasUnitWithinDistanceFromOpponent(N): Checks if the ally player has a unit within a distance N from a opponent's unit.

6. u.hasNumberOfWorkersHarvesting(N): Checks if the ally player has N units of type Worker harvesting resources.

7. u.is_Type(T): Checks if a unit is an instance of type T.

8. u.isBuilder(): Checks if a unit is of type Worker.

9. u.canAttack(): Checks if a unit can attack.

10. u.hasUnitThatKillsInOneAttack(): Checks if the ally player has a unit that kills an opponent's unit with one attack action.

11. u.opponentHasUnitThatKillsUnitInOneAttack(): Checks if the opponent player has a unit that kills an ally's unit with one attack action.

12. u.hasUnitInOpponentRange(): Checks if an unit of the ally player is within attack range of an opponent's unit.

13. u.opponentHasUnitInPlayerRange(): Checks if an unit of the opponent player is within attack range of an ally's unit.

14. u.canHarvest(): Checks if a unit can harvest resources.

The Command functions ('C' in the CFG) are described below:

1. u.build(T, D, N): Builds N units of type T on a cell located on the D direction of the unit.

2. u.train(T, D, N): Trains N units of type T on a cell located on the D direction of the structure responsible for training them.

3. u.moveToUnit(T_p, O_p): Commands a unit to move towards the player T_p following a criterion O_p.

4. u.attack(O_p): Sends N Worker units to harvest resources.

5. u.harvest(N): Sends N Worker units to harvest resources.

6. u.attackIfInRange(): Commands a unit to stay idle and attack if an opponent unit comes within its attack range.

7. u.moveAway(): Commands a unit to move in the opposite direction of the player's base.

'T' represents the types a unit can assume. 'N' is a set of integers. 'D' represents the directions available used in action functions.

'O_p' is a set of criteria to select an opponent unit based on their current state. 'T_p' represents the set of target players.

The following 5 are some guidelines for writing the playing strategy:

1. There is NO NEED TO write classes or initiate objects such as Unit, Worker, etc. There is also NO NEED TO write comments.

2. Use curly braces like C/C++/Java while writing any 'for' or 'if' or 'if-else' block. Start the curly braces in the same line of the block.

3. Do not write 'else if(B) {' block. Write 'else { if(B) {...}}' instead.

4. A strategy must be written inside one or multiple 'for' blocks.

5. You must not use any symbols (for example: &&, ||, etc.) outside the CFG. In case of code like 'if (B1 && B2)', write 'if (B1) { if (B2) {...}}' instead.

Now your tasks are the following 7:

1. Understand the Boolean (B) and command (C) functions from above and try to relate them in the context of MicroRTS playing strategies.

2. Write a program in the MicroRTS language encoding a very strong game-playing strategy for the 9x8 map described above. You must follow the guidelines for writing the playing strategy while writing your program.

3. You must not use any symbols (for example &&, ||, etc.) that the CFG does not accept. You have to strictly follow the CFG while writing the program.

4. Look carefully, the methods of non-terminal symbols B and C have prefixes 'u.' in the examples since they are methods of the object 'Unit u'. You should follow the patterns of the examples.

5. Write only the pseudocode inside '<strategy></strategy>' tag.

6. Do not write unnecessary symbols of the CFG such as, '$S \rightarrow$', '$\rightarrow$', etc.

7. Check the program and ensure it does not violate the rules of the CFG or the guidelines for writing the strategy.

## C.2 First Attempt (Example for 24x24 Map)

Consider a 24x24 gridded map of MicroRTS, a real-time strategy game. Consider this map as a 2 dimensional array with the following structure:

– There is a wall in the middle of the map consisting of two columns that has a small passage of 4 cells. The small passage consists of 4 resource cells each having only 1 resource.

– There are 28 resource cells at the top-left, top-right, bottom-left and bottom-right corners of the map respectively where each of them contains 10 units of resources.

– The bases of player 1 are located at indices (3,2) and (20,2), located on both sides of the wall.

– The bases of player 2 are located at indices (20,21) and (3,21), also located on both sides of the wall.

– Each player controls two bases, which initially have 5 units of resources each.

– There are 2 workers beside each base. So a total of 4 workers for each of the players.

Consider this Context-Free Grammar (CFG) describing a programming language for writing programs encoding strategies of MicroRTS. The CFG is shown in the <CFG></CFG> tag bellow:

$$\begin{aligned}
&\text{<CFG>}\\
&\quad S \rightarrow SS \mid \text{for(Unit u) S} \mid \text{if(B) then S}\\
&\qquad\quad \mid \text{if(B) then S else S} \mid C \mid \lambda\\
&\quad B \rightarrow u.\text{hasNumberOfUnits}(T, N)\\
&\qquad\quad \mid u.\text{opponentHasNumberOfUnits}(T, N)\\
&\qquad\quad \mid u.\text{hasLessNumberOfUnits}(T, N)\\
&\qquad\quad \mid u.\text{haveQtdUnitsAttacking}(N)\\
&\qquad\quad \mid u.\text{hasUnitWithinDistanceFromOpponent}(N)\\
&\qquad\quad \mid u.\text{hasNumberOfWorkersHarvesting}(N)\\
&\qquad\quad \mid u.\text{is\_Type}(T)\\
&\qquad\quad \mid u.\text{isBuilder}()\\
&\qquad\quad \mid u.\text{canAttack}()\\
&\qquad\quad \mid u.\text{hasUnitThatKillsInOneAttack}()\\
&\qquad\quad \mid u.\text{opponentHasUnitThatKillsUnitInOneAttack}()\\
&\qquad\quad \mid u.\text{hasUnitInOpponentRange}()\\
&\qquad\quad \mid u.\text{opponentHasUnitInPlayerRange}()\\
&\qquad\quad \mid u.\text{canHarvest}()\\
&\quad C \rightarrow u.\text{build}(T, D, N) \mid u.\text{train}(T, D, N) \mid u.\text{moveToUnit}(T_p, O_p)\\
&\qquad\quad \mid u.\text{attack}(O_p) \mid u.\text{harvest}(N)\\
&\qquad\quad \mid u.\text{attackIfInRange}() \mid u.\text{moveAway}()\\
&\quad T \rightarrow \text{Base} \mid \text{Barracks} \mid \text{Ranged} \mid \text{Heavy}\\
&\qquad\quad \mid \text{Light} \mid \text{Worker}\\
&\quad N \rightarrow 0 \mid 1 \mid 2 \mid 3 \mid 4 \mid 5 \mid 6 \mid 7 \mid 8 \mid 9\\
&\qquad\quad \mid 10 \mid 15 \mid 20 \mid 25 \mid 50 \mid 100\\
&\quad D \rightarrow \text{EnemyDir} \mid \text{Up} \mid \text{Down} \mid \text{Right} \mid \text{Left}\\
&\quad O_p \rightarrow \text{Strongest} \mid \text{Weakest} \mid \text{Closest} \mid \text{Farthest}\\
&\qquad\quad \mid \text{LessHealthy} \mid \text{MostHealthy} \mid \text{Random}\\
&\quad T_p \rightarrow \text{Ally} \mid \text{Enemy}\\
&\text{</CFG>}
\end{aligned}$$

This language allows nested loops and conditionals. It contains several Boolean functions (B) and command-oriented functions (C) that provide either information about the current state of the game or commands for the ally units.
The Boolean functions ('B' in the CFG) are described below:

1. u.hasNumberOfUnits(T, N): Checks if the ally player has N units of type T.

2. u.opponentHasNumberOfUnits(T, N): Checks if the opponent player has N units of type T.

3. u.hasLessNumberOfUnits(T, N): Checks if the ally player has less than N units of type T.

4. u.haveQtdUnitsAttacking(N): Checks if the ally player has N units attacking the opponent.

5. u.hasUnitWithinDistanceFromOpponent(N): Checks if the ally player has a unit within a distance N from a opponent's unit.

6. u.hasNumberOfWorkersHarvesting(N): Checks if the ally player has N units of type Worker harvesting resources.

7. u.is_Type(T): Checks if a unit is an instance of type T.

8. u.isBuilder(): Checks if a unit is of type Worker.

9. u.canAttack(): Checks if a unit can attack.

10. u.hasUnitThatKillsInOneAttack(): Checks if the ally player has a unit that kills an opponent's unit with one attack action.

11. u.opponentHasUnitThatKillsUnitInOneAttack(): Checks if the opponent player has a unit that kills an ally's unit with one attack action.

12. u.hasUnitInOpponentRange(): Checks if an unit of the ally player is within attack range of an opponent's unit.

13. u.opponentHasUnitInPlayerRange(): Checks if an unit of the opponent player is within attack range of an ally's unit.

14. u.canHarvest(): Checks if a unit can harvest resources.

The Command functions ('C' in the CFG) are described below:

1. u.build(T, D, N): Builds N units of type T on a cell located on the D direction of the unit.

2. u.train(T, D, N): Trains N units of type T on a cell located on the D direction of the structure responsible for training them.

3. u.moveToUnit(T_p, O_p): Commands a unit to move towards the player T_p following a criterion O_p.

4. u.attack(O_p): Sends N Worker units to harvest resources.

5. u.harvest(N): Sends N Worker units to harvest resources.

6. u.attackIfInRange(): Commands a unit to stay idle and attack if an opponent unit comes within its attack range.

7. u.moveAway(): Commands a unit to move in the opposite direction of the player's base.

'T' represents the types a unit can assume. 'N' is a set of integers. 'D' represents the directions available used in action functions.
'O_p' is a set of criteria to select an opponent unit based on their current state. 'T_p' represents the set of target players.

Now consider the following program encoding a strategy for playing MicroRTS written inside '<strategy-1></strategy-1>' tag:
<strategy-1>

```
1  for(Unit u){
2    for(Unit u){
3      u.build(Barracks, Right, 50)
4    }
5    u.train(Worker, Up, 4)
6    u.attack(Strongest)
7    for(Unit u){
8      u.harvest(5)
9    }
10   u.moveToUnit(Ally, Closest)
11   for(Unit u){
12     u.train(Ranged, Up, 5)
13   }
14 }
```
</strategy-1>

Now your tasks are the following 3:

1. Analyze strategy-1 and try to analyze its weaknesses.

2. Write a new strategy that defeats strategy-1.

3. You need to only write this new strategy inside '<counterStrategy></counterStrategy>' tag.

## C.3 Feedback Attempt (Example for 32x32 Map)

Consider a 32x32 map of MicroRTS, a real-time strategy game. Consider this map as a 2 dimensional array with the following structure:

– There are two L-shaped obstacles on the map, each with a passage of 4 cells located at the middle of left and right sides.

– There are a total of 12 neutral resource cells R located at the top-right and bottom-left corners of the map. Each resource center contains 20 units of resources.

– The base B1 of player 1 is located at index (6,14), which is located on the left side of the map.

– The base B2 of player 2 is located at index (25,17), which is located on the right side of the map.

– Each player controls one Base, which initially has 20 units of resources.

– There is one worker for each player besides their bases.

Consider this Context-Free Grammar (CFG) describing a programming language for writing programs encoding strategies of MicroRTS. The CFG is shown in the <CFG></CFG> tag bellow:

$$<\text{CFG}>$$
$$S \rightarrow SS \mid \text{for(Unit u) S} \mid \text{if(B) then S}$$
$$\mid \text{if(B) then S else S} \mid C \mid \lambda$$
$$B \rightarrow u.\text{hasNumberOfUnits}(T, N)$$
$$\mid u.\text{opponentHasNumberOfUnits}(T, N)$$
$$\mid u.\text{hasLessNumberOfUnits}(T, N)$$
$$\mid u.\text{haveQtdUnitsAttacking}(N)$$
$$\mid u.\text{hasUnitWithinDistanceFromOpponent}(N)$$
$$\mid u.\text{hasNumberOfWorkersHarvesting}(N)$$
$$\mid u.\text{is\_Type}(T)$$
$$\mid u.\text{isBuilder}()$$
$$\mid u.\text{canAttack}()$$
$$\mid u.\text{hasUnitThatKillsInOneAttack}()$$
$$\mid u.\text{opponentHasUnitThatKillsUnitInOneAttack}()$$
$$\mid u.\text{hasUnitInOpponentRange}()$$
$$\mid u.\text{opponentHasUnitInPlayerRange}()$$
$$\mid u.\text{canHarvest}()$$
$$C \rightarrow u.\text{build}(T, D, N) \mid u.\text{train}(T, D, N) \mid u.\text{moveToUnit}(T_p, O_p)$$
$$\mid u.\text{attack}(O_p) \mid u.\text{harvest}(N)$$
$$\mid u.\text{attackIfInRange}() \mid u.\text{moveAway}()$$
$$T \rightarrow \text{Base} \mid \text{Barracks} \mid \text{Ranged} \mid \text{Heavy}$$
$$\mid \text{Light} \mid \text{Worker}$$
$$N \rightarrow 0 \mid 1 \mid 2 \mid 3 \mid 4 \mid 5 \mid 6 \mid 7 \mid 8 \mid 9$$
$$\mid 10 \mid 15 \mid 20 \mid 25 \mid 50 \mid 100$$
$$D \rightarrow \text{EnemyDir} \mid \text{Up} \mid \text{Down} \mid \text{Right} \mid \text{Left}$$
$$O_p \rightarrow \text{Strongest} \mid \text{Weakest} \mid \text{Closest} \mid \text{Farthest}$$
$$\mid \text{LessHealthy} \mid \text{MostHealthy} \mid \text{Random}$$
$$T_p \rightarrow \text{Ally} \mid \text{Enemy}$$
$$</\text{CFG}>$$

This language allows nested loops and conditionals. It contains several Boolean functions (B) and command-oriented functions (C) that provide either information about the current state of the game or commands for the ally units.

The Boolean functions ('B' in the CFG) are described below:

1. u.hasNumberOfUnits(T, N): Checks if the ally player has N units of type T.

2. u.opponentHasNumberOfUnits(T, N): Checks if the opponent player has N units of type T.

3. u.hasLessNumberOfUnits(T, N): Checks if the ally player has less than N units of type T.

4. u.haveQtdUnitsAttacking(N): Checks if the ally player has N units attacking the opponent.

5. u.hasUnitWithinDistanceFromOpponent(N): Checks if the ally player has a unit within a distance N from a opponent's unit.

6. u.hasNumberOfWorkersHarvesting(N): Checks if the ally player has N units of type Worker harvesting resources.

7. u.is_Type(T): Checks if a unit is an instance of type T.

8. u.isBuilder(): Checks if a unit is of type Worker.

9. u.canAttack(): Checks if a unit can attack.

10. u.hasUnitThatKillsInOneAttack(): Checks if the ally player has a unit that kills an opponent's unit with one attack action.

11. u.opponentHasUnitThatKillsUnitInOneAttack(): Checks if the opponent player has a unit that kills an ally's unit with one attack action.

12. u.hasUnitInOpponentRange(): Checks if an unit of the ally player is within attack range of an opponent's unit.

13. u.opponentHasUnitInPlayerRange(): Checks if an unit of the opponent player is within attack range of an ally's unit.

14. u.canHarvest(): Checks if a unit can harvest resources.

The Command functions ('C' in the CFG) are described below:

1. u.build(T, D, N): Builds N units of type T on a cell located on the D direction of the unit.

2. u.train(T, D, N): Trains N units of type T on a cell located on the D direction of the structure responsible for training them.

3. u.moveToUnit(T_p, O_p): Commands a unit to move towards the player T_p following a criterion O_p.

4. u.attack(O_p): Sends N Worker units to harvest resources.

5. u.harvest(N): Sends N Worker units to harvest resources.

6. u.attackIfInRange(): Commands a unit to stay idle and attack if an opponent unit comes within its attack range.

7. u.moveAway(): Commands a unit to move in the opposite direction of the player's base.

'T' represents the types a unit can assume. 'N' is a set of integers. 'D' represents the directions available used in action functions.
'O_p' is a set of criteria to select an opponent unit based on their current state. 'T_p' represents the set of target players.

Now consider the following program encoding a strategy for playing MicroRTS written inside '<strategy-1></strategy-1>' tag:
<strategy-1>

```
 1  for(Unit u){
 2    for(Unit u){
 3      u.build(Barracks, Right, 50)
 4    }
 5    u.train(Worker, Up, 4)
 6    u.attack(Strongest)
 7    for(Unit u){
 8      u.harvest(5)
 9    }
10    u.moveToUnit(Ally, Closest)
11    for(Unit u){
```

```
12       u.train(Ranged, Up, 5)
13     }
14   }
```
</strategy-1>

Here is a strategy that could not defeat the above strategy:
<strategy-2>
```
 1  for(Unit u){
 2     for(Unit u){
 3        u.build(Barracks,Left,1)
 4     }
 5     u.train(Heavy,Down,4)
 6     u.train(Worker,Up,4)
 7     u.attack(Strongest)
 8     for(Unit u){
 9        u.harvest(5)
10     }
11     u.moveToUnit(Enemy,Closest)
12     for(Unit u){
13        u.train(Ranged,Up,5)
14     }
15  }
```
</strategy-2>

The following is an encoding of the units, which we will use to give you information about a match played between strategy-1 and strategy-2 above.
Base : B
Worker : W
Ranged : Rg
Light : Li
Heavy : Hv
Barracks : Br

The following is an encoding of the actions:
attack_location : att_loc
return : ret
wait : wt
move : mv
produce : prod
harvest : har

The following is an encoding of the directions:
left : l
right : r
up : u
down : d

The following is a randomly sampled sequence of actions of the match played between strategy-1 as player 0 and strategy-2 as player 1:
{ (B(4)(0, (1,1), 10, 0),prod(u,W)) },
{ (B(6)(1, (7,6), 10, 0),prod(u,W))(W(16)(1, (6,5), 1, 1),mv(u))(W(18)(1, (6,6), 1, 0),mv(l)) },

```
{ (W(17)(0, (0,3), 1, 1),mv(u))(W(19)(0, (1,0), 1, 0),mv(r))(W(21)(0, (2,1), 1,
1),ret(l))(B(4)(0, (1,1), 10, 0),wt(10)) },
{ (W(15)(0, (1,2), 1, 0),mv(d))(W(19)(0, (3,0), 1, 1),mv(l))(W(21)(0, (3,1), 1,
1),mv(l))(B(4)(0, (1,1), 10, 0),wt(10))(W(17)(0, (0,1), 1, 0),wt(10))(Br(23)(0, (2,2),
4, 0),wt(10)) },
{ (W(18)(1, (7,7), 1, 0),mv(l))(W(22)(1, (5,6), 1, 1),mv(r))(Rg(26)(1, (5,4), 1,
0),mv(u))(B(6)(1, (7,6), 10, 0),wt(10))(W(16)(1, (6,5), 1, 0),wt(10))(W(20)(1, (7,5),
1, 0),wt(10))(Br(24)(1, (5,5), 4, 0),wt(10)) },
{ (Rg(25)(0, (3,2), 1, 0),wt(10)) },
{ (Rg(28)(0, (2,5), 1, 0),att_loc(5,5)) },
{ (B(6)(1, (7,6), 10, 0),prod(l,W))(W(33)(1, (7,5), 1, 0),mv(u))(W(37)(1, (7,7), 1,
0),mv(l))(Br(36)(1, (6,5), 4, 0),wt(10)) },
{ (Rg(25)(0, (5,4), 1, 0),att_loc(7,6))(Rg(31)(0, (6,4), 1, 0),att_loc(7,6))(Rg(28)(0,
(3,6), 1, 0),wt(10)) },
{ (B(6)(1, (7,6), 2, 0),wt(10)) }
```
The strategy-2 failed to defeat strategy-1.

Now your tasks are the following 3:

1. Analyze strategy-1 and try to analyze its weaknesses. For this analysis, you may take help from the sequence of actions from the match between strategy-1 and strategy-2 we provided.

2. Write a new strategy that defeats strategy-1.

3. You need to only write this new strategy inside '<counterStrategy></counterStrategy>' tag.

## C.4 Encrypted DSL

Consider this Context-Free Grammar (CFG) describing a programming language for writing programs encoding strategies of MicroRTS. The CFG is shown in the <CFG></CFG> tag bellow:

<CFG>

$S \rightarrow SS \mid$ for(Unit u) S $\mid$ if(B) then S
$\mid$ if(B) then S else S $\mid C \mid \lambda$

$B \rightarrow u.\text{b1}(T, N) \mid u.\text{b2}(T, N)$
$\mid u.\text{b3}(T, N) \mid u.\text{b4}(N)$
$\mid u.\text{b5}(N) \mid u.\text{b6}(N)$
$\mid u.\text{b7}(\text{T}) \mid u.\text{b8}()$
$\mid u.\text{b9}() \mid u.\text{b10}()$
$\mid u.\text{b11}() \mid u.\text{b12}()$
$\mid u.\text{b13}() \mid u.\text{b14}()$

$C \rightarrow u.\text{c1}(T, D, N) \mid u.\text{c2}(T, D, N) \mid u.\text{c3}(T_p, O_p)$
$\mid u.\text{c4}(O_p) \mid u.\text{c5}(N)$
$\mid u.\text{c6}() \mid u.\text{c7}()$

$T \rightarrow \text{t1} \mid \text{t2} \mid \text{t3} \mid \text{t4}$
$\mid \text{t5} \mid \text{t6}$

$N \rightarrow 0 \mid 1 \mid 2 \mid 3 \mid 4 \mid 5 \mid 6 \mid 7 \mid 8 \mid 9$
$\mid 10 \mid 15 \mid 20 \mid 25 \mid 50 \mid 100$

$D \rightarrow \text{d1} \mid \text{d2} \mid \text{d3} \mid \text{d4} \mid \text{d5}$

$O_p \rightarrow \text{op1} \mid \text{op2} \mid \text{op3} \mid \text{op4}$
$\mid \text{op5} \mid \text{op6} \mid \text{op7}$

$T_p \rightarrow \text{tp1} \mid \text{tp2}$

</CFG>

This language allows nested loops and conditionals. It contains several Boolean functions (B) and command-oriented functions (C) that provide either information about the current state of the game or commands for the ally units.

The Boolean functions ('B' in the CFG) are described below:

1. u.b1(T, N): Checks if the ally player has N units of type T.

2. u.b2(T, N): Checks if the opponent player has N units of type T.

3. u.b3(T, N): Checks if the ally player has less than N units of type T.

4. u.b4(N): Checks if the ally player has N units attacking the opponent.

5. u.b5(N): Checks if the ally player has a unit within a distance N from a opponent's unit.

6. u.b6(N): Checks if the ally player has N units of type Worker harvesting resources.

7. u.b7(T): Checks if a unit is an instance of type T.

8. u.b8(): Checks if a unit is of type Worker.

9. u.b9(): Checks if a unit can attack.

10. u.b10(): Checks if the ally player has a unit that kills an opponent's unit with one attack action.

11. u.b11(): Checks if the opponent player has a unit that kills an ally's unit with one attack action.

12. u.b12(): Checks if an unit of the ally player is within attack range of an opponent's unit.

13. u.b13(): Checks if an unit of the opponent player is within attack range of an ally's unit.

14. u.b14(): Checks if a unit can harvest resources.

The Command functions ('C' in the CFG) are described below:

1. u.c1(T, D, N): Builds N units of type T on a cell located on the D direction of the unit.

2. u.c2(T, D, N): Trains N units of type T on a cell located on the D direction of the structure responsible for training them.

3. u.c3(T_p, O_p): Commands a unit to move towards the player T_p following a criterion O_p.

4. u.c4(O_p): Sends N Worker units to harvest resources.

5. u.c5(N): Sends N Worker units to harvest resources.

6. u.c6(): Commands a unit to stay idle and attack if an opponent unit comes within its attack range.

7. u.c7(): Commands a unit to move in the opposite direction of the player's base.

'T' represents the types of units as the following:

1. t1: Base

2. t2: Barracks

3. t3: Ranged

4. t4: Heavy

5. t5: Light

6. t6: Worker

'D' represents directions as the following:

1. d1: EnemyDir

2. d2: Up

3. d3: Down

4. d4: Right

5. d5: Left

'O_p' is a set of criteria to select an opponent unit based on its current state like the following:

1. op1: Strongest

2. op2: Weakest

3. op3: Closest

4. op4: Farthest

5. op5: LessHealthy

6. op6: MostHealthy

'T_p' represents the set of target players like the following:

1. tp1: Ally

2. tp2: Enemy

Finally, 'N' is a set of integers.

# D   Poachers and Rangers Prompts

## D.1   Initial Attempt

We have an environment called 'Poachers and Rangers' where 2 teams called poachers and rangers are competing with each other in a national park and its surroundings. The park has 60 gates in total. The goal for each team is to defeat the opponents.

Now I have the following CFG to write programs for poachers in the above environment:

$$S \to SA \mid A$$
$$A \to \text{attack(n)}$$
$$n \to 1 \mid 2 \mid 3 \mid \ldots \mid 59 \mid 60$$

The following is the CFG to write programs for rangers:

$$S \to SA \mid A$$
$$A \to \text{defend(n)}$$
$$n \to 1 \mid 2 \mid 3 \mid \ldots \mid 59 \mid 60$$

The following is the explanation of the above CFG:
CFG Explanation:

S: Starting symbol that can contain one or multiple actions.

A: Refers to the action taken by the team.

attack(n): Refers to the action to attack the n-th gate of the park

defend(n): Refers to the action to defend the n-th gate of the park

n: Any positive integer up to 60.

...: It is not part of the CFG. It has been used to indicate all positive numbers in between.

The following are some guidelines for writing the strategy program:

1. There is NO NEED TO write classes or initiate objects such as Poachers, Rangers, Gates, etc. There is NO NEED TO write comments.

2. DO NOT write '...' in the program, since it is not a part of the CFG.

3. Write only the action or the sequence of actions such as 'attack(a)' or 'attack(a) attack(b) attack(c)' where a, b and c are positive integers.

Now your tasks are the following 5:

1. Understand all the symbols of the given CFG in the context of this given 'Poachers and Rangers' environment.

2. Write a very strong strategy for poachers/rangers considering the given environment, the above CFG and the strategy writing guidelines.

3. You must not use any symbols (for example '$S \to$', '$\to$', '$\mid$', etc.) outside the given CFG. Write only the action or the sequence of action as mentioned in the strategy writing guideline.

4. Replace '..' with contents meant by this symbol if there are any, then write only the strategy program inside '<strategy></strategy>' tag.

5. Check the strategy program and ensure it does not violate the rules of the CFG or the guidelines for writing the strategy.

## D.2 First Attempt

We have an environment called 'Poachers and Rangers' where 2 teams called poachers and rangers are competing with each other in a national park and its surroundings. The park has 60 gates in total. The goal for each team is to defeat the opponents.

Now I have the following CFG to write programs for poachers in the above environment:

$$S \to SA \mid A$$
$$A \to \text{attack(n)}$$
$$n \to 1 \mid 2 \mid 3 \mid \ldots \mid 59 \mid 60$$

The following is the CFG to write programs for rangers:

$$S \to SA \mid A$$
$$A \to \text{defend(n)}$$
$$n \to 1 \mid 2 \mid 3 \mid \ldots \mid 59 \mid 60$$

The following is the explanation of the above CFG:

S: Starting symbol that can contain one or multiple actions.

A: Refers to the action taken by the team.

attack(n): Refers to the action to attack the n-th gate of the park

defend(n): Refers to the action to defend the n-th gate of the park

n: Any positive integer up to 60.

...: It is not part of the CFG. It has been used to indicate all positive numbers in between.

The following are some guidelines for writing the strategy program:

1. There is NO NEED TO write classes or initiate objects such as Poachers, Rangers, Gates, etc. There is NO NEED TO write comments.

2. DO NOT write '...' in the program, since it is not a part of the CFG.

3. Write only the action or the sequence of actions such as 'attack(a)' or 'attack(a) attack(b) attack(c)' where a, b and c are positive integers.

Now I have the following strategy program for the poachers that satisfies the CFG, written inside '<strategy-1></strategy-1>' tag:
<strategy-1>
attack(1)
</strategy-1>

Now your tasks are the following 5:

1. Understand all the symbols of the given CFG in the context of this given 'Poachers and Rangers' environment.

2. Write an improved strategy for rangers that can defeat strategy-1.

3. You must not use any symbols (for example '$S \rightarrow$', '$\rightarrow$', '|', etc.) outside the given CFG. Write only the action or the sequence of action as mentioned in the strategy writing guideline.

4. Replace '...' with contents meant by this symbol if there are any, then write only the new strategy program inside the '<rangersStrategy></rangersStrategy>' tag.

5. Check the strategy program and ensure it does not violate the rules of the CFG or the guidelines for writing the strategy.

## D.3 Feedback Attempt

We have an environment called 'Poachers and Rangers' where 2 teams called poachers and rangers are competing with each other in a national park and its surroundings. The park has 60 gates in total. The goal for each team is to defeat the opponents.

Now I have the following CFG to write programs for poachers in the above environment:

$$S \rightarrow SA \mid A$$
$$A \rightarrow \text{attack}(n)$$
$$n \rightarrow 1 \mid 2 \mid 3 \mid \ldots \mid 59 \mid 60$$

The following is the CFG to write programs for rangers:

$$S \rightarrow SA \mid A$$
$$A \rightarrow \text{defend}(n)$$
$$n \rightarrow 1 \mid 2 \mid 3 \mid \ldots \mid 59 \mid 60$$

The following is the explanation of the above CFG:

S: Starting symbol that can contain one or multiple actions.

A: Refers to the action taken by the team.

attack(n): Refers to the action to attack the n-th gate of the park

defend(n): Refers to the action to defend the n-th gate of the park

n: Any positive integer up to 60.

...: It is not part of the CFG. It has been used to indicate all positive numbers in between.

The following are some guidelines for writing the strategy program:

1. There is NO NEED TO write classes or initiate objects such as Poachers, Rangers, Gates, etc. There is NO NEED TO write comments.

2. DO NOT write '...' in the program, since it is not a part of the CFG.

3. Write only the action or the sequence of actions such as 'attack(a)' or 'attack(a) attack(b) attack(c)' where a, b and c are positive integers.

Now I have the following strategy program for the poachers that satisfies the CFG, written inside '<strategy-1></strategy-1>' tag:
<strategy-1>
attack(1)
</strategy-1>

Here is a strategy written inside <strategy2></strategy2> tag for rangers that failed to defeat the given poachers' strategy1:
<strategy-2>
defend(60)
</strategy-2>

The rangers following this strategy-2 defended 1 gate(s), but could not defend gate 1, whereas, the poachers following the strategy-1 attacked gate 1.
Now your tasks are the following 6:

1. Understand all the symbols of the given CFG in the context of this given 'Poachers and Rangers' environment.

2. Analyze why strategy-2 could not defeat strategy-1.

3. Write an improved strategy-2 for rangers that can defeat strategy-1.

4. You must not use any symbols (for example '$S \rightarrow$', '$\rightarrow$', '|', etc.) outside the given CFG. Write only the action or the sequence of action as mentioned in the strategy writing guideline.

5. Replace '...' with contents meant by this symbol if there are any, then write only the new strategy program inside the '<rangersStrategy></rangersStrategy>' tag.

6. Check the strategy program and ensure it does not violate the rules of the CFG or the guidelines for writing the strategy.

# E  Climbing Monkey Prompts

## E.1  Initial Attempt

We have an environment called 'Climbing Monkey' where 2 monkeys are competing with each other to climb a tree. The tree has an infinite number of branches. The goal for one monkey is to defeat another monkey. Now I have the following CFG to write programs for the above environment:

$$S \rightarrow SA \mid A$$
$$A \rightarrow \text{climb}(n)$$
$$n \rightarrow 1 \mid 2 \mid 3 \mid \ldots \mid \text{infinity}$$

The following is the explanation of the above CFG:

S: Starting symbol that can contain one or multiple actions

A: Refers to the action taken by the monkey.

climb(n): Refers to the action to climb the n-th branch of the tree.

n: Any positive integer upto infinity.

...: It is not part of the CFG. It has been used to indicate all positive numbers in between.

The following are some guidelines for writing the strategy program:

1. There is NO NEED TO write classes or initiate objects such as Monkey, Tree, etc. There is NO NEED TO write comments.

2. DO NOT write '...' in the program, since it is not a part of the CFG.

3. Write only the action or the sequence of actions such as 'climb(a)' or 'climb(a) climb(b) climb(c)' where a, b and c are positive integers.

Now your tasks are the following 5:

1. Understand all the symbols of the given CFG in the context of this given 'Climbing Monkey' environment.

2. Write a very strong strategy considering the given environment, the above CFG and the strategy writing guidelines.

3. You must not use any symbols (for example '$S \to$', '$\to$', '$|$', '...' etc.) outside the given CFG. Remove these symbols if there are any.

4. Replace '...' with contents meant by this symbol if there are any, then write only the strategy program inside '<strategy></strategy>' tag.

5. Check the strategy program and ensure it does not violate the rules of the CFG or the guidelines for writing the strategy.

## E.2 First Attempt

We have an environment called 'Climbing Monkey' where 2 monkeys are competing with each other to climb a tree. The tree has an infinite number of branches. The goal for one monkey is to defeat another monkey.

Now I have the following CFG to write programs for the above environment:

$$S \to SA \mid A$$
$$A \to \text{climb(n)}$$
$$n \to 1 \mid 2 \mid 3 \mid \ldots \mid \text{infinity}$$

The following is the explanation of the above CFG:

S: Starting symbol that can contain one or multiple actions

A: Refers to the action taken by the monkey.

climb(n): Refers to the action to climb the n-th branch of the tree.

n: Any positive integer upto infinity.

...: It is not part of the CFG. It has been used to indicate all positive numbers in between.

The following are some guidelines for writing the strategy program:

1. There is NO NEED TO write classes or initiate objects such as Monkey, Tree, etc. There is NO NEED TO write comments.

2. DO NOT write '. . .' in the program, since it is not a part of the CFG.

3. Write only the action or the sequence of actions such as 'climb(a)' or 'climb(a) climb(b) climb(c)' where a, b and c are positive integers.

Now I have the following strategy program that satisfies the CFG, written inside '<strategy-1></strategy-1>' tag:
<strategy-1>
climb(1) climb(2) climb(3)
</strategy-1>

Now your tasks are the following 5:

1. Understand all the symbols of the given CFG in the context of this given 'Climbing Monkey' environment.

2. Write an improved strategy that will help the monkey defeat another monkey following strategy-1.

3. You must not use any symbols (for example '$S \rightarrow$', '$\rightarrow$', '|', etc.) outside the given CFG. Write only the action or the sequence of action as mentioned in the strategy writing guideline.

4. Replace '. . .' with contents meant by this symbol if there are any, then write only the new strategy program inside the '<newStrategy></newStrategy>' tag.

5. Check the strategy program and ensure it does not violate the rules of the CFG or the guidelines for writing the strategy.

### E.3 Feedback Attempt

We have an environment called 'Climbing Monkey' where 2 monkeys are competing with each other to climb a tree. The tree has an infinite number of branches. The goal for one monkey is to defeat another monkey.

Now I have the following CFG to write programs for the above environment:

$$S \rightarrow SA \mid A$$
$$A \rightarrow \text{climb(n)}$$
$$n \rightarrow 1 \mid 2 \mid 3 \mid \dots \mid \text{infinity}$$

The following is the explanation of the above CFG:

S: Starting symbol that can contain one or multiple actions

A: Refers to the action taken by the monkey.

climb(n): Refers to the action to climb the n-th branch of the tree.

n: Any positive integer upto infinity.

. . . : It is not part of the CFG. It has been used to indicate all positive numbers in between.

The following are some guidelines for writing the strategy program:

1. There is NO NEED TO write classes or initiate objects such as Monkey, Tree, etc. There is NO NEED TO write comments.

2. DO NOT write '...' in the program, since it is not a part of the CFG.

3. Write only the action or the sequence of actions such as 'climb(a)' or 'climb(a) climb(b) climb(c)' where a, b and c are positive integers.

Now I have the following strategy program that satisfies the CFG, written inside '<strategy-1></strategy-1>' tag:
<strategy-1>
climb(1) climb(2) climb(3)
</strategy-1>

Here is a strategy that could not improve the given strategy written inside '<strategy-2></strategy-2>' tag:
<strategy-2>
climb(1) climb(2) climb(4) climb(8) climb(16)
</strategy-2>
The monkey following strategy2 could climb 2 branches, whereas, the opponent monkey following strategy1 could climb 3 branches.

Now your tasks are the following 6:

1. Understand all the symbols of the given CFG in the context of this given 'Climbing Monkey' environment.

2. Analyze why strategy-2 above could not improve strategy-1.

3. Write an improved strategy-2 that will help the monkey defeat another monkey following strategy-1.

4. You must not use any symbols (for example '$S \rightarrow$', '$\rightarrow$', '|', etc.) outside the given CFG. Write only the action or the sequence of action as mentioned in the strategy writing guideline.

5. Replace '...' with contents meant by this symbol if there are any, then write only the new strategy program inside the '<newStrategy></newStrategy>' tag.

6. Check the strategy program and ensure it does not violate the rules of the CFG or the guidelines for writing the strategy.

