# OpenReview forum: "Language Models Speed Up Local Search for Finding Programmatic Policies"
_TMLR — Accepted by TMLR_

### Review · Reviewer_zNMB · 2024-08-16

**Summary Of Contributions:**

This paper addresses the problem of improving the sample efficiency of methods which synthesize reinforcement learning (RL) policies described in a domain-specific language. This work presents Local Search with LLM (LS-LLM), which utilizes the programming skills and general knowledge of large language models (LLMs) to produce initial program candidates, and then further improves these programs using a local search method, e.g., hill climbing. The experiments on various grid-world tasks, e.g., Poachers & Rangers, Climbing Monkey, and MicroRTS, show that the proposed framework significantly outperforms local search methods without LLM-initialized programs. I believe this work studies a promising research direction, presents an interesting framework, and provides sufficient evaluations. In sum, I recommend accepting this paper since it presents valuable contributions that would interest the community.

**Audience:**

Yes

**Claims And Evidence:**

Yes

**Requested Changes:**

Please respond to my questions above.

**Strengths And Weaknesses:**

## Paper strengths and contributions

**Motivation and intuition**
- The motivation for utilizing the programming skills and general knowledge of LLMs to improve the sample efficiency of synthesizing program policies is convincing.

**Clarity**
- The overall writing is clear. The authors utilize figures well to illustrate the ideas.

**Related work**
- The authors sufficiently discuss the related work.

**Experimental results**
- The experimental results show that the proposed framework significantly outperforms local search methods.

## Paper weaknesses and questions

**Minor issues**
- Typos
    - "LS-LLMbut" the last paragraph of the introduction
- SHC is not defined. I am assuming it means Stochastic Hill Climbing.

**Deep RL baselines**
- It would be interesting to see how the proposed framework and the local search methods compared to naive deep RL methods, such as PPO or SAC.

**Other programmatic baselines**
- As mentioned in the introduction, existing works have explored learning RL policies structured in decision trees or finite state machines. It would be informative if a comparison to these methods, or an explanation of why comparing these methods is non-trivial, were provided.

**Other search methods**
- This paper focuses on SHC. It would be interesting to see how other search methods perform.

**Elo**
The winning rate is used as the evaluation metric. I wonder if it would make sense to use Elo.

**Generalizability**
- As stated in the introduction, programs can generalize to unseen scenarios better than neural network policies. Evaluating such generalizability would strengthen this work.

---

> ### Author Response · Authors · 2024-09-03
>
> We thank the reviewer for spending time reading and giving great suggestions for our work.
>
> >"LS-LLMbut" the last paragraph of the introduction
>
> Fixed.
>
> > SHC is not defined. I am assuming it means Stochastic Hill Climbing.
>
> Fixed.
>
> > Deep RL baselines
> > It would be interesting to see how the proposed framework and the local search methods compared to naive deep RL methods, such as PPO or SAC.
>
> Thank you for this suggestion. We had considered running this comparison before submitting the paper. However, we decided against it for the following reasons. First, training Deep RL agents for MicroRTS requires a non-trivial amount of compute. Second, last year's MicroRTS competition already compared Programmatic RL and Deep RL. The Deep RL method RAISocketAI requires a substantial amount of compute to learn how to play the game, and yet it loses to 2L, the Programmatic RL baseline we use in our study, which we improve upon.
>
> We have added the following paragraph in Section 5.4 to explain this decision:
>
> *We did not consider deep reinforcement learning baselines because we focus on testing our hypothesis that LLMs can be used to speed up the synthesis of programmatic policies. Moreover, the 2023 MicroRTS competition showed the general trend that one should expect from such a comparison. On smaller maps, a neural policy outperforms programmatic ones. This is because the DSL used in the competition and also in our experiments does not allow for fine-grained control of the game units, which neural policies can quickly learn on smaller maps. However, as the size of the maps increases, programmatic policies outperform neural ones. This is because the inductive bias the DSL provides allows for the synthesis of policies with strong high-level strategies (e.g., how to train stronger units and reach the opponent in larger spaces). See the MicroRTS AI Competition (2023) website for more information.*
>
> > Other programmatic baselines
> > As mentioned in the introduction, existing works have explored learning RL policies structured in decision trees or finite state machines. It would be informative if a comparison to these methods, or an explanation of why comparing these methods is non-trivial, were provided.
>
> This is a good point. We have added the following paragraph in the revised version (see the paragraph in blue in Section 5.4) justifying our decision.
>
> *We also did not consider other programmatic representations such as decision trees (Bastani et al., 2018) and finite state machines (Koul et al., 2019) in our experiments because they require one to first train a neural model that is then distilled into a programmatic representation or to directly map the neural model onto a program (Orfanos & Lelis, 2023). Due to this dependency on using a neural policy to guide the synthesis process, it is unclear how to leverage LLMs to speed up learning with these representations. One could use decision trees and finite state machines, without neural guidance, as the underlying representation of the policies. However, we would lose the inductive bias the domain-specific language provides while still paying the cost of having to search in discrete and non-differentiable spaces of programs.*
>
> > Other search methods
> > This paper focuses on SHC. It would be interesting to see how other search methods perform.
>
> We chose SHC because it is possibly the simplest method and was shown to be a strong algorithm in previous work. Moreover, previous work suggested that the local search algorithm is not as important as the representation used to define the search space (Carvalho et al. 2024).
>
> > Elo The winning rate is used as the evaluation metric. I wonder if it would make sense to use Elo.
>
> That is an interesting idea and comparing different metrics could be worthwhile in future research. We used winning rate because it is the standard metric used in the related literature.
>
> > Generalizability
> > As stated in the introduction, programs can generalize to unseen scenarios better than neural network policies. Evaluating such generalizability would strengthen this work.
>
> This is a good suggestion. The reason we did not perform this experiment is that we wanted to focus on the hypothesis that LLMs can speed up the synthesis process. Note that previous work showed that program reuse in MicroRTS can be done through parts of programs and not necessarily with the programs as a whole (Moraes and Lelis, 2024).
>
> **References**
>
> Tales Henrique Carvalho, Kenneth Tjhia, and Levi H. S. Lelis. Reclaiming the source of programmatic policies: Programmatic versus latent spaces. In The Twelfth International Conference on Learning Representations, 2024.
>
> Rubens O. Moraes and Levi H. S. Lelis. Searching for programmatic policies in semantic spaces. 2024

---

> > ### Comment · Reviewer_zNMB · 2024-09-04
> > **Re: Official Comment by Authors**
> >
> > I appreciate the revision and the responses from the authors. My concerns are sufficiently addressed. I recommend accepting this paper as it presents solid contributions, exploring utilizing the programming skills and general knowledge of large language models (LLMs) to produce initial program candidates, and then further improving these programs using a local search method. I believe the findings would interest a wide range of TMLR audiences.

---

> > > ### Author Response · Authors · 2024-09-06
> > > **Thank you!**
> > >
> > > Thank you for the quick response on our rebuttal and for being supportive of our work.

---

### Review · Reviewer_uSqq · 2024-08-28

**Summary Of Contributions:**

This paper uses LLMs to seed the local search algorithm for program search. It is a straightforward idea that shows good improvements on the used environments, especially for Poachers & Rangers and Climbing Monkey. For MicroRTS the improvements are also good but not that significant as for the two other environments, which makes sense when you look on the environment and the used domain-specific language.

**Audience:**

Yes

**Broader Impact Concerns:**

I don't see the need to add an broader impact statement.

**Claims And Evidence:**

Yes

**Requested Changes:**

Currently, it is quite difficult for me to say what should be changed to make this work acceptable, as the claim about the training data of GPT 3.5 for MicroRTS is wrong. I would appreciate a response from the authors, how they would rewrite their claims regarding this new information.

**Strengths And Weaknesses:**

The paper is written clearly and easy to understand.

Great improvements for the Poachers & Rangers and Climbing Monkey environments, but on the other hand, I don't think these are very representative DSLs for program search since no control flow structures are used, only attack, defend, climb, and integer functions are available. Also, the problem domain is pretty easy to understand, so it makes sense that the LLM is so much better for these two environments. However, it's still impressive to see.

It would be interesting to see the same ablations for the other two environments as well to backup the claim of the abstract that “LLMs are effective in this setting because we give them access to the outcomes of rollouts of the policies”.

When reading the prompts, I also have the feeling that a lot of prompt engineering was necessary to make the algorithm work. Also example strategies are provided for the LLM. Does it make a difference when you seed the 2L search algorithm with the same strategy provided to the LLM? Because currently the search algorithms without an LLM are randomly initialized.

Some claims for MicroRTS in the paper are incorrect. See this part in the paper: “We used GPT 3.5 which was trained with data up to January 2022. To the best of our knowledge, no programmatic policy for PR, CM, and MicroRTS had been published online before the model’s training cut-off date.”

There are strategies available on github which were provided with the first paper when introducing the DSL as well. See for example here: https://github.com/julianmarino/LS2/blob/master/LS2_AAAI/TableInitialPortfolio/ScriptsTable.txt (June 8 2021)

and here https://github.com/julianmarino/mRTS_Compiler_LS2/tree/master/Compiler_mRTS_LS2_AAAI/src/ai/synthesis/grammar (April 19 2021)

The paper states that GPT 3.5 is used with data until January 2022 but if I look on the OpenAI site it only shows GPT 3.5 turbo with data up to September 2021. Is it not possible to specify a specific version you used? Otherwise https://platform.openai.com/docs/models/gpt-3-5-turbo

Both repositories where avaiable before the cutoff date of the training data of GPT 3.5.

No source code is provided for reproducibiliy.

Regarding this it is quite hard to evaluate the usefullnes of the LLM with the MicroRTS environment since the training data for GPT 3.5 is contaminated with data about it. Additionally the Microlanguage looks a lot like Java, which is the same language of all the code related to the MicroRTS environment. (which is not really a weakness of the paper but would explain the usefullness of an LLM for this DSL)

---

> ### Author Response · Authors · 2024-09-03
>
> Thank you for your detailed review and suggestions. Please see our comments below.
>
> > It would be interesting to see the same ablations for the other two environments as well to backup the claim of the abstract that “LLMs are effective in this setting because we give them access to the outcomes of rollouts of the policies”.
>
> This is a good point and something we discussed among ourselves before submitting the paper. As the reviewer noted, the other two domains are easy for the LLM, so we thought that such an ablation would not be as informative as the ablation on the much more difficult MicroRTS. Thus we decided to save on API cost by skipping this ablation on the easier domains.
>
> > When reading the prompts, I also have the feeling that a lot of prompt engineering was necessary to make the algorithm work.
>
> This is difficult for us to quantify because we did not measure how much time we spent writing and modifying the prompt. However, the effort spent on prompt writing was to certify obvious constraints. For example, the prompt we wrote helps the LLM follow the syntax and structure of the language; it also helps the LLM generate the program in a way that our system can copy and use it (the XML tags). The prompt was not tuned to produce stronger programs. In the future, we expect that the effort to make an LLM follow the precise syntax of a DSL will be minimized through methods that appropriately constrain sampling from the LLM (see e.g. Brandon T. Willard, Rémi Louf (2023). “Efficient Guided Generation for Large Language Models”)
>
> > Also example strategies are provided for the LLM. Does it make a difference when you seed the 2L search algorithm with the same strategy provided to the LLM? Because currently the search algorithms without an LLM are randomly initialized.
>
> This is an important point we failed to explain in our submission. The program we provide in the prompt is the target strategy for which the LLM needs to compute a best response. Recall that self-play algorithms compute best responses to a sequence of policies (see Section 4.1). The baselines we use also leverage this information as they seed their search with the latest policy computed in the self-play process. To answer your question, the algorithms without the LLM do not start with a randomly initialized program, but with possibly a fairly complex one, computed in the last iteration of self-play (i.e., same policy used in the prompt). We added the following sentences at the end of the first paragraph of Section 5.4 of the revised version to clarify this.
>
> Note that in self-play algorithms, we need to compute a sequence of best-responses---i.e., solve a sequence of MDPs. While 2L(LS-LLM) uses an LLM to initialize each of these searches, the baselines 2L(LS), FP(LS), and IBR(LS) use the latest best response computed to initialize the search in the programmatic space.

---

> > ### Author Response · Authors · 2024-09-03
> >
> > > Some claims for MicroRTS in the paper are incorrect. See this part in the paper: “We used GPT 3.5 which was trained with data up to January 2022. To the best of our knowledge, no programmatic policy for PR, CM, and MicroRTS had been published online before the model’s training cut-off date.”
> >
> > > There are strategies available on github which were provided with the first paper when introducing the DSL as well. See for example here: https://github.com/julianmarino/LS2/blob/master/LS2_AAAI/TableInitialPortfolio/ScriptsTable.txt (June 8 2021)
> >
> > > and here https://github.com/julianmarino/mRTS_Compiler_LS2/tree/master/Compiler_mRTS_LS2_AAAI/src/ai/synthesis/grammar (April 19 2021)
> >
> > Thank you for bringing this to our attention. The programs shown in the paper and in the repository of this 2021 paper by Mariño et al. were written in a different language. Although both languages are called Microlanguage, the version we used has departed significantly from the original one. Our interpreter failed to run all the programs listed in the links above. To illustrate some differences, the instruction moveToUnit(Light, Ally, strongest, u) in the older Microlanguage has four parameters, while in ours it only has two. The older language has a larger collection of high-level functions that are often used by the synthesizer, such as HaveQtdUnitsbyType. In addition to this, we used 2 maps that were not used by Mariño et al. In MicroRTS, different maps might require entirely different programs to play well.
> >
> > Considering this, we have updated our text to the following.
> >
> > *We used GPT 3.5, which was trained with data up to September 2021. To the best of our knowledge, no programmatic policy for PR, CM, and MicroRTS had been published online before the model’s training cut-off date. The repository of Mariño et al. (2021) contains programs written in an earlier version of Microlanguage. This earlier version of the language differs significantly from the version we use in our experiments in terms of the domain-specific functions available and the number of parameters required in the functions. To illustrate, none of the programs available in Mariño et al.’s repository can be run in the interpreter of our version of the language. Moreover, Mariño et al. did not experiment with the NoWhereToRun and BWDistantResources maps that we use. Considering the differences in the language and in the maps used, it is unlikely that Mariño et al.’s programs have influenced our results.*
> >
> > > The paper states that GPT 3.5 is used with data until January 2022 but if I look on the OpenAI site it only shows GPT 3.5 turbo with data up to September 2021. Is it not possible to specify a specific version you used? Otherwise https://platform.openai.com/docs/models/gpt-3-5-turbo
> >
> > Well spotted. The cut-off date is indeed September 2021. We have fixed that in the paper.
> >
> > > Both repositories where avaiable before the cutoff date of the training data of GPT 3.5.
> >
> > Please see our answer above.
> >
> > > No source code is provided for reproducibility.
> >
> > We will make our code available after the reviewing process.
> >
> > > Regarding this it is quite hard to evaluate the usefulness of the LLM with the MicroRTS environment since the training data for GPT 3.5 is contaminated with data about it. Additionally the Microlanguage looks a lot like Java, which is the same language of all the code related to the MicroRTS environment. (which is not really a weakness of the paper but would explain the usefullness of an LLM for this DSL)
> >
> > We have clarified the issue related to the data available online. Please let us know if you have further questions. We agree that the similarity of the Microlanguage to general-purpose languages such as Java is what makes this a promising research direction. We can leverage the LLM's knowledge to speed up the process of learning programmatic policies.
> >
> > > Requested Changes:
> > > Currently, it is quite difficult for me to say what should be changed to make this work acceptable, as the claim about the training data of GPT 3.5 for MicroRTS is wrong. I would appreciate a response from the authors, how they would rewrite their claims regarding this new information.
> >
> > Thank you again for bringing this to our attention. We hope our answer has clarified this issue.

---

> > > ### Comment · Reviewer_uSqq · 2024-09-21
> > >
> > > Thank you for clarifying my points. I think they have been sufficiently addressed. I changed the checkbox for "Claims and Evidence" to Yes. So I would now also recommend for accepting the paper.

---

### Review · Reviewer_2wVK · 2024-09-17

**Summary Of Contributions:**

The authors propose utilizing large language models (LLMs) to enhance traditional search algorithms for generating programmatic policies. The general idea is to generate an initial policy in the form of an abstract syntax tree using an LLM, which is then further optimized using a search algorithm. The proposed algorithm LS-LLM goes a step further by first allowing the LLM to iteratively optimize its initial policy, using feedback from the previous runs. Once the LLM optimization has been completed, the best-performing policy is optimized using the traditional search algorithm. The authors apply this algorithm to multiple two-player zero-sum games using self-play. In this scenario, LS-LLM is repeatedly used to beat the previous best-performing policy.

**Audience:**

Yes

**Broader Impact Concerns:**

No concerns

**Claims And Evidence:**

Yes

**Requested Changes:**

- 5.4 Baseline Systems: The first sentence contains a double "of of" (Not critical)
- Experiments: An analysis of generated strategies in MicroRTS would help better support your hypothesis that the LLM explores different kinds of strategies (Not critical)
- A figure for visualizing LS-LLM with self-play would help tremendously to understand how it works. I had quite some trouble figuring out if and when the LLM is used to generate a new policy (Not critical)

**Strengths And Weaknesses:**

The presented paper contains strong results supporting the author's goal to show that LLMs speed up search algorithms for programmatic policies. The conducted experiments clearly show that LS-LLM far outperforms traditional algorithms. The ablation experiments also show that LS-LLM performs much worse when not providing the LLM with feedback for improving its initial policy.

It is a bit unclear how much impact the traditional search algorithm has. The authors mention that in MicroRTS, the LLM often fails to generate optimal policies, which are easily found by the traditional search algorithms. They hypothesize that the LLM still helps improve results by generating different kinds of policies, thus enabling much more efficient exploration of the search space. While I can see how this makes sense, a more thorough examination of the generated types of policies would have helped support their claim.

In the ablation experiments, the authors also show that by obfuscating some of the names used in the abstract syntax tree, the results become much worse. While this is an interesting observation, it doesn't seem to add much to the claim of the paper.

To my knowledge, the approach presented in this paper is novel, and the experiments show strong results.

---

> ### Author Response · Authors · 2024-10-10
>
> Thank you for your review and detailed comments and suggestions.
>
> We have uploaded a revision of the paper where we made the following changes.
>
> > 5.4 Baseline Systems: The first sentence contains a double "of of" (Not critical)
>
> Fixed.
>
> > Experiments: An analysis of generated strategies in MicroRTS would help better support your hypothesis that the LLM explores different kinds of strategies (Not critical)
>
> We added two representative examples of policies that the LLM generated to the same target policy provided by the self-play algorithm. These examples highlight how the LLM allows the search to quickly explore different types of strategies in the game. Please see Appendix B of the revised paper.
>
> > A figure for visualizing LS-LLM with self-play would help tremendously to understand how it works. I had quite some trouble figuring out if and when the LLM is used to generate a new policy (Not critical)
>
> We added a schematic view of LS-LLM, as suggested. Please see Figure 2 on page 4 of the revised paper.

---

### Decision · Action_Editor_nCzV · 2024-10-08

**Recommendation:** Accept as is

**Comment:**

Following from the discussions between authors and reviewers, and the final recommendations provided to me by the reviewers, I have no further requirements for revision and recommend to accept the paper as is.

Of course, preparing the camera-ready version will still allow for minor changes, and I would like to recommend / encourage the authors to remember and take into account the following:
- Some reviewers asked about sharing source code. If you are able to, please do this and include a URL for the code repository.
- Reviewer 2wVK (the reviewer with whom you did not yet actively discuss, who I think submitted their review after you last revised the paper) had some additional suggestions / requests. I would like to encourage you to still consider these (certainly fixing the typo, but you may also wish to consider the request for including some sort of figure that provides a general overview of the proposed algorithm).

**Audience:**

Using combinations of search and LLMs for program synthesis is clearly within the scope of TMLR and of interest to its audience.

**Claims And Evidence:**

Reviewers agree that this paper has extensive and appropriate experiments, including strong results and ablations, which provide proper support for the claims and conclusions.

---

> ### Author Response · Authors · 2024-10-10
>
> Dear Action Editor,
>
> Thank you for your message and recommendation on our submission. We have addressed the concerns of Reviewer 2wVK and submitted a revision of our work. We will now prepare the camera-ready version and include the link to our repository with the implementation used in our experiments.
>
> Thank you for your work handling our paper.
>
> Best,
> Authors